

**The role of dust mineral composition in atmospheric radiation and pollution in North China: new insights from EMIT and two-way coupled modeling**

Chao Gao[1], Xuelei Zhang[1,2,*], Hu Yang[1], Ling Huang[3], Hongmei Zhao[1], Shichun Zhang[1], and Aijun Xiu[1]

[1]State Key Laboratory of Black Soils Conservation and Utilization, Northeast Institute of Geography and Agroecology, Chinese Academy of Sciences, Changchun, 130102, China

[2]School of Geographical Sciences, Liaoning Normal University, Dalian, 116029, China

[3]School of Environmental and Chemical Engineering, Shanghai University, Shanghai 200444, China

[*]Correspondence to: X.L. Zhang (zhangxuelei@neigae.ac.cn)

**Abstract**

Mineral dust is a major atmospheric aerosol influencing Earth's energy balance through aerosol-radiation (ARI) and aerosol-cloud interactions (ACI). While homogeneous dust effects have been studied, the impact of mineralogical composition on regional meteorology and air quality remains underexplored, limiting accurate forecasting of dust storm impacts, especially in dust belt regions. In this study, we used a two-way coupled WRF-CHIMERE model with three mineralogical dust atlases (Nickovic et al. (2012) (N2012), Journet et al. (2014) (J2014), and a new dataset, Li et al. (2024) (L2024), from the Earth Surface Mineral Dust Source Investigation (EMIT)) to evaluate ARI effects during the March 2021 dust storm in North China. Results showed significant spatial variations in radiative forcing due to mineralogical differences. Bulk dust (without considering mineralogy) caused an average shortwave radiative forcing of $-5.72$ W/m², while mineral-specific forcings increased this by up to $+0.10$ W/m². Integrating EMIT data reduced $PM_{10}$ biases by over 15% in high-concentration regions and improved ozone predictions, with localized changes of $-2.46$ to $+3.52$ μg/m³. Hematite's strong absorption and quartz's reflective properties were key in altering radiative and air quality outcomes. Compared to scenarios of bulk dust, the consideration of ARI effects of mineralogical compositions can increase $PM_{10}$ concentration by up to 1189.48 μg/m³ in dust source regions. Future research perspectives on the utilization of high-resolution EMIT data in two-way coupled meteorology and air quality models for investigating the ACI effects of





mineralogical dust on cloud microphysics are proposed.

## 1 Introduction

Mineral dust, a dominant component of global atmospheric aerosols, primarily
originates from wind erosion in arid and semi-arid regions (Schepanski, 2018; Shao et al.,
2011). It can affect the Earth's energy balance through direct scattering and absorption of
solar, i.e. aerosol-radiation interaction (ARI), as well as indirect effects on cloud properties
by acting as cloud condensation nuclei and ice nuclei, i.e., aerosol-cloud interaction (ACI)
(Choobari et al., 2014; Kok et al., 2023). By altering biogeochemical cycles, atmospheric
chemistry and visibility, and air quality, transported mineral dust can exacerbate economic
losses, and health risks (Adebiyi et al., 2023; Cwiertny et al., 2008; Duniway et al., 2019;
Maher et al., 2010; Tong et al., 2023). Accurate forecasting of mineral dust events is crucial
to mitigate these adverse impacts.
Numerous studies have demonstrated that the magnitude of dust ARI and ACI effects
is significantly influenced by its mineralogical composition. For instance, iron oxides,
particularly hematite and goethite, have been identified as key components responsible for
dust absorption of solar radiation, as evidenced by both observational and modeling studies
(Alfaro et al., 2004; Gómez Maqueo Anaya et al., 2024; Lafon et al., 2006; Li et al., 2022;
Obiso et al., 2024; Scanza et al., 2015; Song et al., 2024). Concurrently, a growing body of
research has explored the impact of various dust mineral compositions, including hematite,
corundum, kaolinite, muscovite, montmorillonite, quartz, calcite, illite, amorphous silicon,
aluminum silicate, and potassium feldspar, on ice nucleation processes. Among these,
potassium feldspar has emerged as a crucial component for dust nucleation activation
(Harrison et al., 2016; Kumar et al., 2018). However, a notable gap exists in our
understanding of how specific mineral compositions impact meteorology and air quality
through ARI and ACI effects. Prior research has predominantly focused on homogeneous
dust aerosols, assuming globally uniform composition and optical properties. However,
this assumption introduces regional inaccuracies in estimating the impacts of dust aerosols,



which remain poorly understood due to uncertainties in dust composition (Ke et al., 2022; Klingmüller et al., 2019; Kok et al., 2017).

Many efforts have been directed to improve simulations of dust mineralogy and its representation in numerical models (Balkanski et al., 2021; Gómez Maqueo Anaya et al., 2024; Gonçalves Ageitos et al., 2023; Li et al., 2021, 2022, 2024; Li and Sokolik, 2018; Menut et al., 2020; Obiso et al., 2024; Scanza et al., 2015; Solomos et al., 2023b, a; Song et al., 2024). Most of the above are offline models, with only two studies conducting two-way feedback simulations with only WRF-Chem (Li and Sokolik, 2018) and WRF-CHIMERE (Menut et al., 2020) being applied. However, both of these studies are derived from artificially generated data and lack effective ground-based validation, as discussed in Claquin et al. (1999), Nickovic et al. (2012) (N2012 hereafter), and Journet et al. (2014) (J2014 hereafter). These validations predominantly focus on agricultural regions rather than the arid and semi-arid areas that are major sources of dust emissions (Green et al., 2020). EMIT instrument provides a new approach to invert and obtain the surface soil mineral composition and further assess the ARI and ACI effects of dust minerals (Connelly et al., 2021). To the best of our knowledge, no prior research has investigated the impact of dust on regional meteorology and air quality while considering its mineral speciation using two-way coupled models with three different mineralogical dust atlases.

Due to the nucleation mechanism of aeolian mineral components (ACI effect) not being incorporated into the two-way coupled model, this study utilizes a two-way coupled WRF-CHIMERE model incorporating three mineral databases to explore how dust particle mineral composition interacts with radiation and North China during a heavy dust storm event. The following sections detail the methodology (model configurations and data sources) in Section 2, analyze the WRF-CHIMERE simulations focusing on ARI effects of dust mineral composition on meteorology and air quality in Section 3, and summarize the key findings in Section 4.





## 2 Methodology and data

### 2.1 Model configurations and data sources

The two-way coupled WRF model version 3.7.1 and CHIMERE model version 2020r3 were employed to simulate the ARI and ACI effects of mineralogical dust particles on meteorology and air quality over North China from March 12 to March 15, 2021, as shown in Fig. A1. The exchanges between meteorological and air quality variable are accomplished through the OASIS coupler (Briant et al., 2017). The simulation was conducted at a horizontal resolution of 27 km, with 165 grid cells in the east-west direction and 87 in the north-south direction, and the study domain is depicted in Fig. A1. The model has 33 vertical levels from surface to 50 hPa with 13 layers in the bottom 1km and the bottom thickness being 24.5 m. The Rapid Radiative Transfer Model for General circulation models (RRTMG) shortwave and longwave radiation schemes were employed to investigate the ARI effects (Briant et al., 2017). Additionally, the Thompson cloud microphysics scheme was utilized to assess the impacts of ACI (Tuccella et al., 2019). The initial and boundary conditions (ICs and BCs) for non-dust aerosols are prescribed by the LDMZ-INCA model, while those for dust aerosols are determined by the GOCART model. The options of other physics and chemistry schemes are presented in Table A1. The dry depositions are treated as described in Zhang et al. (2001). The parameterizations for the removal of dust particles below clouds by raindrops and snow are based on the methods proposed by Willis & Tattelman (1989) and Wang et al. (2014), respectively. Inline mineral dust emissions, incorporating mineralogy, are computed using a u* threshold and a dust production model for saltation (Kok et al., 2014; Shao and Lu, 2000). The model accounts for the impact of soil moisture on suppressing mineral dust emissions (Fécan et al., 1998).

For the calculations of ARI effects in WRF-CHIMERE, refractive indices corresponding to these mineralogical species are provided in Table 2 of Menut et al. (2020). Concerning shortwave (SW) radiation, the aerosol optical properties, encompassing single scattering albedos and asymmetry factors at 400 and 600 nm, as well as the aerosol optical depth (AOD) at 300, 400, and 999 nm, calculated using Fast-JX, were interpolated or



extrapolated to obtain values at 14 SW intervals (Briant et al., 2017; Gao et al., 2022). AOD
at 16 longwave (LW) intervals ranging from 3400 to 55600 nm are directly used to
calculate LW radiation.

To evaluate the model performance of the WRF-CHIMERE model with and without

including mineralogical dust emissions, we collected 132 hourly national environmental
observations for $PM_{2.5}$ and $PM_{10}$ concentrations at https://quotsoft.net/air/. The shortwave
radiation (SSR) data, consisting of 59 hourly surface measurements, was sourced from
Tang et al. (2019). The 844 hourly surface meteorology station data can be accessed at
https://data.cma.cn.

**2.2 Mineral dust atlases**

Detailed soil composition data is crucial for separating the emission flux into

contributions from individual minerals. Mineral density and refractive index data are
referenced from Menut et al. (2020). Several global mineralogical composition databases
(N2012, J2014, EMIT) provide data on 12 mineral species (Table 1) at varying resolutions
(1 km × 1 km, 0.5° × 0.5°). These data are interpolated to match the study's model grid.
For consistency, the N2012 data (originally at 1 km × 1 km, accessible at
http://www.seevccc.rs/GMINER30) is resampled to 0.5° × 0.5°. The J2014 data,
implemented in the WRF-CHIMERE model, includes 12 mineral compositions found in
clay and/or silt fractions (details in Table 2 of Menut et al., 2020). EMIT data
(https://earth.jpl.nasa.gov/emit/data/data-products) requires specific processing. Since it
provides only normalized spectral abundance, it needs recalculation to represent the
normalized mass proportion of each mineral in each substrate. Additionally, EMIT lacks
data for feldspar and quartz. In cases where the total EMIT composition suggests less than
100% (indicating a missing mineral contribution), the missing fraction is filled to represent
the relative amounts of quartz and feldspar. As EMIT only combines illite and muscovite,
their individual contents are estimated based on the respective proportions in N2012 or
J2014 data. Regarding specific mineralogical species appear in both clay and silt soil



fractions, the contents of these species for EMIT dataset are calculated based on contents
of respective species in clay and silt soil for N2012 or J2014 datasets. The detailed
calculation steps can be found at https://earth.jpl.nasa.gov/emit/internal_resources/284.

Table 1. Mineralogical compositions in different datasets.

| Mineral | Clay | | Silt | | EMIT |
|---|---|---|---|---|---|
| | N2012 | J2014 | N2012 | J2014 | |
| Smectite | ✓ | ✓ | ✗ | ✗ | ✓ |
| Illite | ✓ | ✓ | ✗ | ✗ | ✓[†] |
| Hematite | ✓ | ✓ | ✓ | ✗ | ✓ |
| Feldspar | ✗ | ✓ | ✓ | ✓ | ✗ |
| Kaolinite | ✓ | ✓ | ✗ | ✗ | ✓ |
| Calcite | ✓ | ✓ | ✓ | ✓ | ✓ |
| Quartz | ✓ | ✓ | ✓ | ✓ | ✗ |
| Gypsum | ✗ | ✗ | ✓ | ✓ | ✓ |
| Vermiculite | ✗ | ✓ | ✗ | ✗ | ✓ |
| Chlorite | ✗ | ✓ | ✗ | ✓ | ✓ |
| Goethite | ✗ | ✗ | ✗ | ✓ | ✓ |
| Mica | ✗ | ✗ | ✗ | ✓ | ✓[†] |
| Resolution | 1km | 0.5° | 1km | 0.5° | 0.5° |

[†] indicates the content of illite + muscovite

**2.3 Scenario set up**
Ten parallel WRF-CHIMERE simulations were performed to investigate the influence
of mineralogical dust on meteorology and air quality in China, employing three distinct
mineralogical atlases, as illustrated in Figs. 1 and A5-A7. Each simulation was conducted
both with and without enabling ARI effects, as detailed in Table 2, to isolate and compare
the effects of mineralogical dust under different modeling conditions.
Simulations without ARI effects (Dust_NO, N2012_default_NO, N2012_EMIT_NO,
J2014_default_NO, J2014_EMIT_NO) were specifically designed to identify the direct
impact of mineralogical dust on meteorology and air quality, independent of the radiative



feedbacks induced by aerosols. These No_ARI simulations served as a baseline for
assessing how mineralogical compositions affect meteorology and air quality in the
absence of aerosol-radiation feedback mechanisms.
In contrast, simulations with ARI enabled (Dust_ARI, N2012_default_ARI,
N2012_EMIT_ARI, J2014_default_ARI, J2014_EMIT_ARI) were used to quantify the
additional effects arising from aerosol-radiation interactions. By comparing simulations
with and without ARI for each mineralogical atlas (e.g., N2012_default_ARI −
N2012_default_NO), the differential impact of ARI effects on meteorology and air quality
for various dust compositions could be identified. This approach highlights how
mineralogical properties of dust influence the strength and nature of ARI effects, thereby
modulating key atmospheric processes such as radiation balance, temperature profiles, and
pollutant distributions.
To evaluate the discrepancies in ARI effects among the mineralogical atlases,
differences in the ARI impacts between EMIT-derived and default dust compositions were
analyzed for both N2012 and J2014 datasets. For example, comparisons such as
(N2012_EMIT_ARI − N2012_EMIT_NO) versus (N2012_default_ARI −
N2012_default_NO) provide insight into the extent to which higher-resolution, satellite-
derived mineralogical data influence ARI effects relative to default atlas-based
representations. Similar comparisons were performed for the J2014 dataset.



Table 2. Summary of dust emission scenarios and aerosol feedback configurations for
different simulation settings.

| Scenario | Emission | Online choice | Coupling type | Aerosol feedback |
|---|---|---|---|---|
| Dust_NO | Bulk dust + anthropogenic emissions | [online] = 1 | [cpl_case] = 1 | No feedbacks |
| Dust_ARI | | [online] = 1 | [cpl_case] = 2 | ARI effects |
| N2012_default_NO | N2012_default dust + anthropogenic emissions | [online] = 1 | [cpl_case] = 1 | No feedbacks |
| N2012_default_ARI | | [online] = 1 | [cpl_case] = 2 | ARI effects |
| N2012_EMIT_NO | N2012_EMIT dust + anthropogenic emissions | [online] = 1 | [cpl_case] = 1 | No feedbacks |
| N2012_EMIT_ARI | | [online] = 1 | [cpl_case] = 2 | ARI effects |
| J2014_default_NO | J2014_default dust + anthropogenic emissions | [online] = 1 | [cpl_case] = 1 | No feedbacks |
| J2014_default_ARI | | [online] = 1 | [cpl_case] = 2 | ARI effects |
| J2014_EMIT_NO | J2014_EMIT dust + anthropogenic emissions | [online] = 1 | [cpl_case] = 1 | No feedbacks |
| J2014_EMIT_ARI | | [online] = 1 | [cpl_case] = 2 | ARI effects |


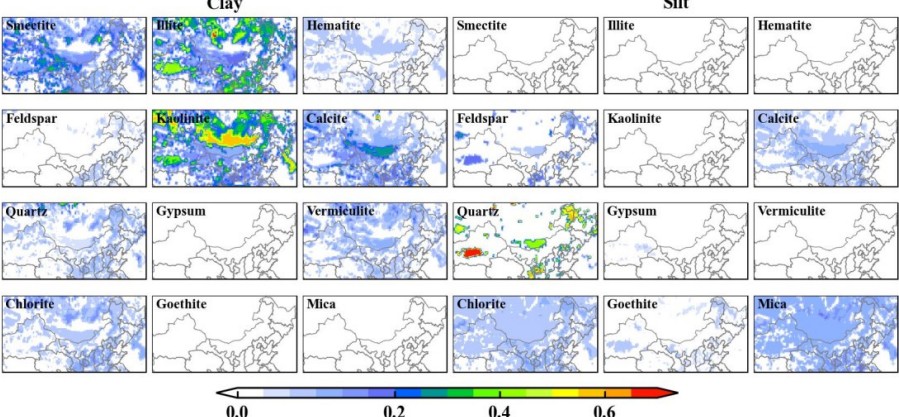


Figure 1. Spatial distribution of content for the different mineral dust species in the silt and clay fraction
of the soil for original J2014 mineralogical data.

**3    Results and discussion**
**3.1 Evaluation of meteorology and air quality**

Table 3 presents the evaluation results for observed and simulated surface shortwave

radiation (SSR), 2-meter temperature (T2), and 10-meter wind speed (WS10) from various
scenario simulations conducted using the WRF-CHIMERE modeling system. The model
exhibits strong overall performance, with correlation coefficients (R) between observed
and simulated values reaching approximately 0.7 for SSR and WS10, and up to 0.93 for



T2. These results demonstrate the model's capacity to capture key atmospheric patterns and
variability across the simulation domain. However, systematic biases are evident,
particularly in North China, where the model tends to overestimate SSR and WS10 by
60.69%–68.92% and 17.06%–17.52%, respectively, while underestimating T2 by 0.48%–
0.58%. These biases suggest challenges in accurately simulating surface radiation fluxes,
near-surface wind dynamics, and temperature fields, especially in areas influenced by high
aerosol concentrations.

A comparative analysis of the two configurations, N2012 and J2014, reveals that

WRF-CHIMERE with N2012 generally outperforms J2014 in simulating SSR and T2,
suggesting that the N2012 parameterization better captures radiative and thermodynamic
processes in the region. Conversely, J2014 exhibits improved accuracy in simulating WS10,
implying potential strengths in its representation of near-surface wind dynamics. These
findings highlight the sensitivity of model performance to parameterization schemes and
the need for tailored configurations for specific meteorological variables. The inclusion of
detailed dust mineralogical compositions, while informative for certain applications,
introduces additional complexities that reduce the overall accuracy of simulations.
Specifically, while these compositions help mitigate the overestimation of SSR and the
underestimation of T2, they exacerbate the overestimation of WS10. The integration of
EMIT satellite data provides a significant boost to model performance, highlighting the
value of incorporating high-resolution, real-time observational data to refine the simulation
of atmospheric variables. EMIT data, with its detailed characterization of aerosol and dust
properties, reduces the positive biases in SSR and WS10 while simultaneously minimizing
the negative biases in T2.

When comparing the ARI (aerosol-radiation interaction) effects of the defaulted

mineralogical compositions in N2012 and J2014 with simulations that implement EMIT
satellite data, the latter shows a clear advantage. Incorporating EMIT data further reduces
the positive biases in SSR and WS10, while simultaneously minimizing the negative biases




in T2. This suggests that EMIT data provides a more precise representation of dust
properties and atmospheric conditions, enhancing the overall reliability of the model.

Table 3. Statistics analysis of daily averaged SSR, 2-meter temperature (T2) and 10-meter wind speed
(WS10) from different scenario simulations and ground observations in North China including
correlation coefficient (R) and normalized mean bias (NMB).

| Scenario | SSR | | T2 | | WS10 | |
|---|---|---|---|---|---|---|
| | R | NMB | R | NMB | R | NMB |
| Dust_NO | 0.7041 | 68.92 | 0.9327 | −0.5816 | 0.7112 | 17.0623 |
| Dust_ARI | 0.7170 | 60.69 | 0.9372 | −0.4831 | 0.7178 | 17.4558 |
| N2012_default_NO | 0.7041 | 68.92 | 0.9327 | −0.5816 | 0.7112 | 17.0623 |
| N2012_default_ARI | 0.7147 | 61.80 | 0.9369 | −0.4758 | 0.7170 | 17.5280 |
| N2012_EMIT_NO | 0.7041 | 68.92 | 0.9327 | −0.5816 | 0.7112 | 17.0623 |
| N2012_EMIT_ARI | 0.7161 | 60.88 | 0.9367 | −0.4799 | 0.7174 | 17.4403 |
| J2014_default_NO | 0.7041 | 68.92 | 0.9327 | −0.5816 | 0.7112 | 17.0623 |
| J2014_default_ARI | 0.7148 | 61.68 | 0.9368 | −0.4779 | 0.7170 | 17.5096 |
| J2014_EMIT_NO | 0.7041 | 68.92 | 0.9327 | −0.5816 | 0.7112 | 17.0623 |
| J2014_EMIT_ARI | 0.7154 | 61.22 | 0.9367 | −0.4796 | 0.7174 | 17.4791 |


To assess the ability of each scenario simulation to replicate regional $PM_{10}$ and $O_3$

temporal patterns, Figure 2 presents hourly time series of simulated and in situ $PM_{10}$ and
$O_3$ concentrations at four North China sites: Ordos, Kalgan, Beijing, and Tianjin. These
locations represent key dust aerosol transport pathways, which play a crucial role in the
region's air quality dynamics due to frequent dust storms and anthropogenic emissions. The
time series plots regarding $PM_{10}$ and $O_3$ in Figures 2 and A2 allow for a direct comparison
of model simulations with observed data, revealing important insights into model
performance across different environmental conditions and geographical settings,
respectively. All models accurately captured the peak $PM_{10}$ and $O_3$ concentrations observed
during the March 12 event in North China, which was characterized by significant dust
emissions and high pollutant levels. This event serves as a key test case for evaluating the
models' responsiveness to extreme atmospheric conditions. However, despite the overall
agreement in peak concentration timings, simulations often overestimated $O_3$ and
underestimated $PM_{10}$ at sites with high dust loads, such as Ordos and Kalgan. This



discrepancy highlights the challenge of simulating the complex interactions between dust
aerosols, precursor gases, and photochemical reactions, particularly in regions with high
dust deposition and frequent air pollution episodes. Additionally, the models tended to
extend the period of elevated $PM_{10}$ concentrations beyond the observed time frame,
suggesting that the processes controlling dust aerosol removal or dispersion were not fully
captured. CHIMERE simulations using J2014 mineralogical data generally outperformed
those using J2012 data, with significant reductions in $PM_{10}$ negative normalized mean
biases (NMBs) for three of the four cities, indicating the importance of accurate
mineralogical characterization of dust for improving model predictions. When considering
the aerosol-radiation interaction (ARI) effects of bulk dust aerosols, $PM_{10}$ negative biases
decreased, and $O_3$ positive biases increased, which suggests that incorporating ARI effects
helps to better represent the impact of dust on local radiative forcing and air quality.
Moreover, incorporating ARI effects from the default dust mineralogical atlas further
enhanced these trends, underscoring the need for refined aerosol property data in enhancing
model performance. Finally, using Earth-observing systems such as the EMIT satellite data
led to substantial reductions in $PM_{10}$ negative bias at Kalgan, Beijing, and Tianjin,
demonstrating the value of remote sensing data in improving model accuracy, particularly
for regions with high aerosol concentrations and complex emission sources.

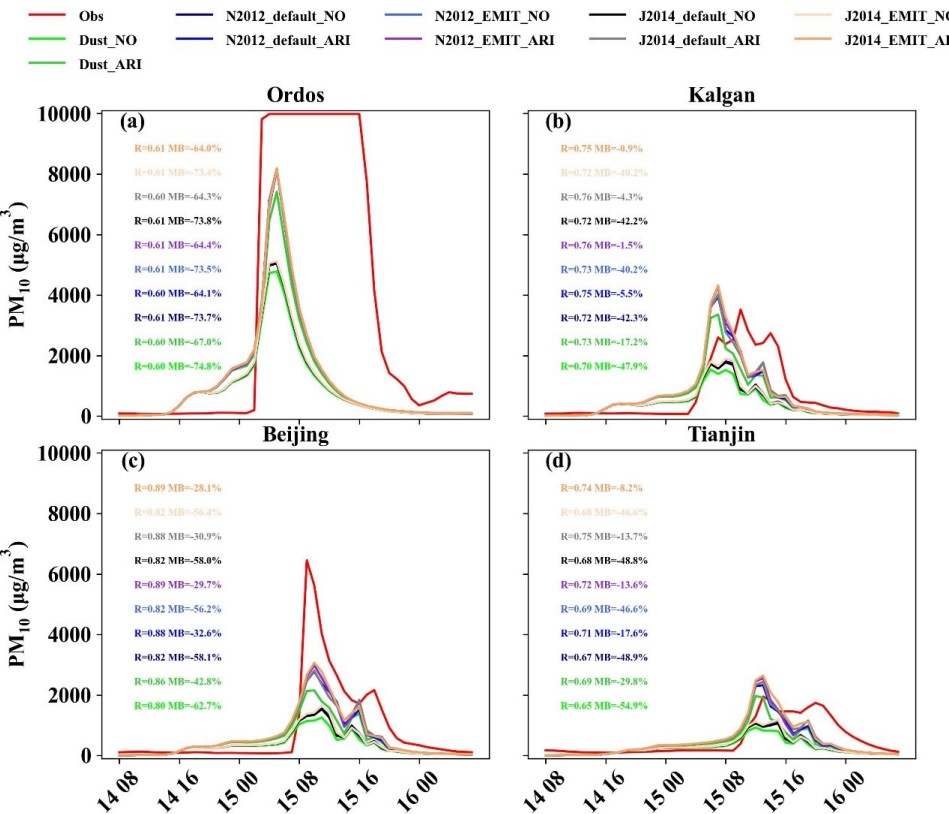

Figure 2. Statatiscal metrices between observed and simulated PM$_{10}$ concentrations by different scenario simulations.

To evaluate the model performance in simulating the horizontal distribution and vertical profile of dust aerosol, Figure 3 presents the false RGB imagery of dust derived from Himawari-8 thermal infrared imagery, along with CALIPSO cross sections of 532 nm total attenuated backscatter and the vertical feature mask for the overpass of China. The figure also includes the corresponding spatial distributions of PM$_{10}$ concentrations at 05:00 UTC on 15th March 2021, a time of significant dust transport in the region. This detailed comparison allows for a comprehensive assessment of how well the model captures both the horizontal and vertical characteristics of dust aerosol distribution. All six experiments show similar dust locations in the atmosphere, which are consistent with the Himawari-8





and CALIPSO observations, suggesting that the models effectively replicate the general
spatial patterns of dust transport. Specifically, the false RGB imagery from Himawari-8
clearly indicates the presence of dust plumes in the atmosphere, with distinct thermal
contrasts that help identify the dust layers. The CALIPSO data, which provide vertical
profiles of aerosol backscatter, further validate the model's ability to capture the vertical
extent and concentration of dust layers. These observations are critical for understanding
the atmospheric processes governing dust dispersion and their impact on air quality. The
close agreement between model simulations and satellite data across all six experiments
also underscores the robustness of the model in representing dust aerosol distribution under
different simulation conditions. This evaluation demonstrates that the models are capable
of simulating the main features of dust aerosol transport, though further refinement in
capturing the fine-scale variations and aerosol properties may still be necessary for more
accurate predictions in future studies.

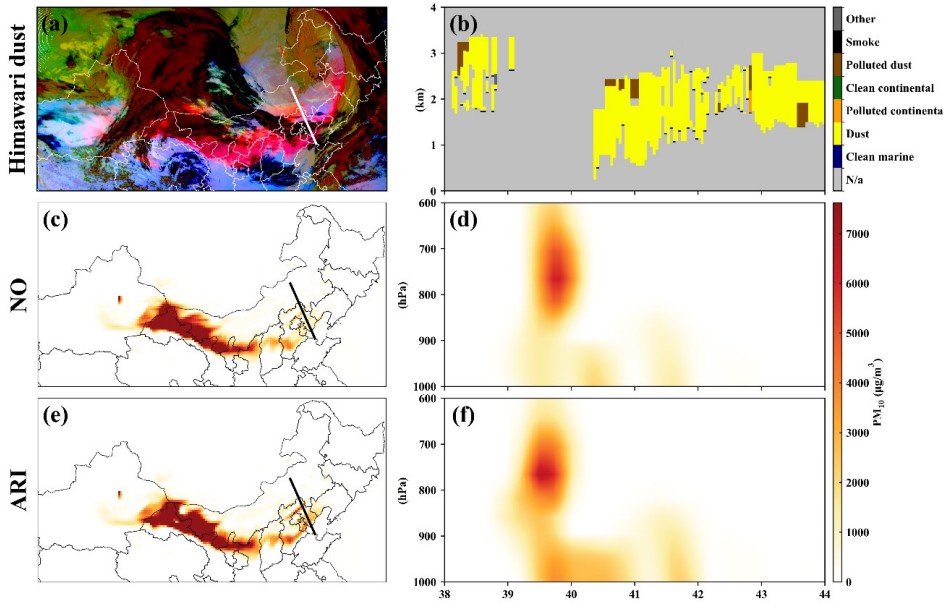


Figure 3. The false color imagery of dust from Himawari-8 thermal infrared imagery(a), CALIPSO
cross sections of 532 nm total attenuated backscatter and the vertical feature mask for the overpass of
China on 13:00 (local time) 15th March 2021 (b), and corresponding horizontal (c and e) and vertical (d
and f) distributions of $PM_{10}$ concentrations.



## 3.2 Radiative effect on meteorology

To further investigate the impacts of dust radiation on energy balance, the spatial distributions of the average shortwave (SW), longwave (LW), and net (NET) radiative forcing induced by bulk dust on the surface (SFC), in the atmosphere (ATM), and at the top of the atmosphere (TOA) are presented during the dust episode shown in Figure 4. The radiative forcing values provide critical insights into the energy exchanges between dust aerosols and the atmosphere, and their subsequent effects on regional climate dynamics. For SW radiation forcings, dust aerosols produced cooling effects at all three layers: the surface, the atmosphere, and the top of the atmosphere. The average SW radiative forcing was about $-5.72$ W m$^{-2}$ at the surface, $-8.69$ W m$^{-2}$ in the atmosphere, and $-2.97$ W m$^{-2}$ at the TOA, highlighting the significant reduction in solar radiation reaching these layers due to the scattering and absorption properties of the dust particles. Particularly in the dust source regions, the cooling effect at the surface exceeded $-900$ W m$^{-2}$ (Figures 4a, 4d, and 4g), indicating the strong influence of dust on the regional energy budget in these areas. This is a result of the large dust concentrations and their optical properties, which effectively block solar radiation from reaching the Earth's surface. In contrast, the dust-induced LW radiative forcing warmed the surface and atmosphere, with average values ranging from 5.78 to 5.86 W m$^{-2}$. This warming effect is associated with the absorption of longwave radiation by dust particles, which then re-radiate heat, contributing to local warming. However, dust particles induced negative LW radiative forcing at the TOA, with values ranging from $-461.88$ to $-379.95$ W m$^{-2}$, reflecting the downward flux of longwave radiation absorbed by the aerosols, which reduces the amount of energy reaching the TOA. The NET radiative forcing, which represents the combined effect of both SW and LW forcings, was positive at the surface (about $+0.15$ W m$^{-2}$), negative in the atmosphere (about $-2.91$ W m$^{-2}$), and negative at the TOA (about $-3.06$ W m$^{-2}$), as shown in Figures 4c, 4f, and 4i. The positive NET radiative forcing at the surface suggests a slight net warming effect at ground level, while the negative values in the atmosphere and at the TOA indicate an overall cooling effect at these higher altitudes.




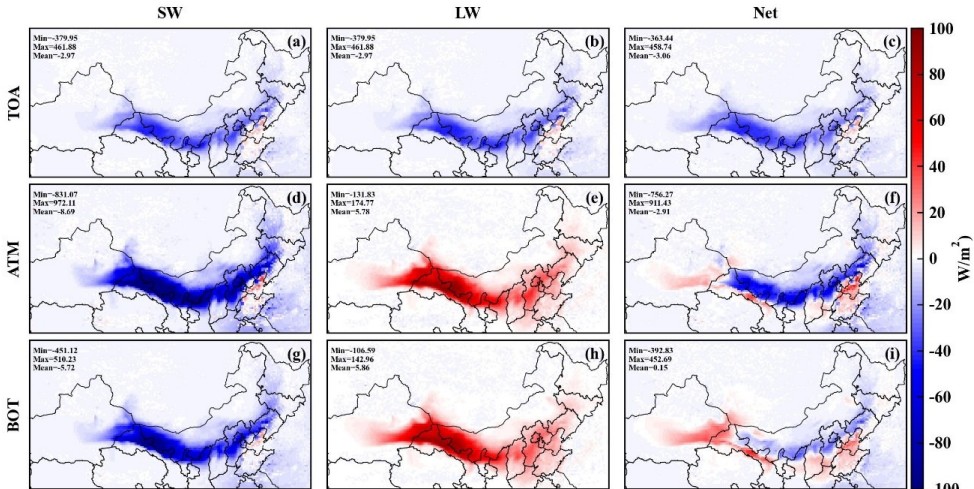


Figure 4. Radiation forcings due to bulk dust enabling ARI effects.

To assess the impact of dust mineralogical composition on radiative forcings, Figure
S6 illustrates the spatial distribution of radiative differences, considering the ARI effects
of bulk dust and comparing them to the default N2012 mineralogy atlas. This comparison
provides valuable insights into how variations in the mineralogical composition of dust
particles can influence the energy balance in the atmosphere. Compared to the ARI effects
of bulk dust, the mineralogical composition of dust aerosols can lead to increases in SW
radiation forcings at the surface and in the atmosphere, ranging from +0.10 to +0.82 W m$^{-2}$.
This increase reflects the different optical properties of dust mineral types, which can affect
the scattering and absorption of solar radiation. These variations in the SW radiation
forcings are particularly important for understanding how different dust types modulate the
amount of solar radiation reaching the Earth's surface and atmosphere. At the TOA,
however, the mineralogical composition resulted in a decrease of about −0.72 W m$^{-2}$ in
SW radiation forcing, suggesting that certain mineralogical types may be more efficient at
reflecting solar radiation back into space. Similar to SW radiation forcings, net radiation
forcings at the surface and in the atmosphere increased, ranging from +0.02 to +0.63 W



m$^{-2}$, while at the TOA, net radiation forcings decreased by about $-0.65$ W m$^{-2}$. The increase
in net radiation at the surface and in the atmosphere reflects the combined effect of
increased SW absorption and the potential changes in longwave (LW) radiative properties.
For LW radiation forcings, the mineralogical composition of dust led to decreases in the
radiative forcing across different layers, ranging from $-0.72$ to $-0.12$ W m$^{-2}$. This decrease
suggests that certain dust mineral types are more efficient at absorbing and emitting
longwave radiation, which can contribute to cooling effects in the atmosphere and at the
surface.

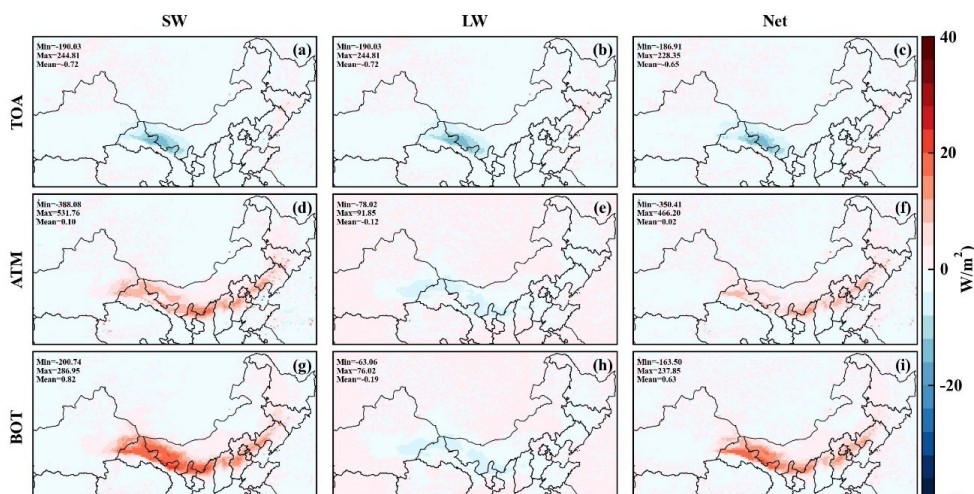


Figure 5. Difference between TOA, ATM and BOT radiation forcings with considering bulk dust and
mineralogical dust compositions (i.e., N2012_default) enabling ARI effects.

As demonstrated in Figure A3, the selection of the soil mineralogy dataset and the

modeling approach significantly influences the calculated dust radiative forcings. When
comparing shortwave dust radiative effects (DRE) from WRF-CHIMERE simulations
using the default N2012 and J2014 mineral atlases, we observe a minor discrepancy in the
DRE amplitude, particularly for shortwave and net radiation forcing at the surface. This
discrepancy suggests that the choice of mineralogical dataset can influence the magnitude
of radiative forcings, especially under varying atmospheric conditions. Previous research



has highlighted the distinct optical properties of hematite and goethite in the shortwave
spectrum (Lafon et al., 2006; Sokolik and Toon, 1999). These differences contribute to
variations in the dust's radiative properties and, in turn, its effect on energy transfer in the
atmosphere. Incorporating both minerals in dust production results in a flatter spectral
single scattering albedo (SSA), as goethite's less pronounced dependence on shortwave
wavelengths reduces the overall absorption in the shortwave spectrum (Formenti et al.,
2014). This effect is particularly noticeable when comparing the radiative forcings from
the different mineralogy datasets, as the presence of goethite alters the absorption and
scattering characteristics of the dust particles.

As depicted in Figure 6, the distinct day-night variations in shortwave radiation

forcing (SWRF) induced by ARI effects have been thoroughly demonstrated when
considering different mineralogical atlases compared to bulk dust. These variations reflect
the different impacts that dust aerosols have on solar radiation during the day and night,
with a clear difference in the magnitude of the effects between the two periods. Notably,
SWRF variations were more pronounced during the daytime than at night, which can be
attributed to the stronger interaction between dust aerosols and incoming solar radiation
during daylight hours. The presence of dust aerosols alters the reflection, absorption, and
scattering of sunlight, leading to significant changes in the radiation balance, especially
during the day when solar energy is at its peak.

Incorporating default dust mineralogical compositions into the simulations led to an

increase in daytime SWRF at the surface and within the atmosphere, ranging from 1.60 to
3.74 W m$^{-2}$. This increase suggests that the specific mineralogy of dust aerosols contributes
to greater absorption and scattering of solar radiation, amplifying the cooling effect at the
surface and the atmosphere. However, at the top of the atmosphere (TOA), the SWRF
decreased by approximately 2.00 W m$^{-2}$, which could be indicative of increased reflection
of shortwave radiation back into space due to the dust particles' optical properties. This
shift in radiative forcing at the TOA highlights the role of dust in altering the energy fluxes



across different atmospheric layers.
When comparing simulations using default dust mineralogical compositions to those
employing Earth-observing EMIT satellite data within the WRF-CHIMERE model,
notable differences in SWRF were observed. Daytime SWRF at the surface was reduced
for the N2012 mineralogy dataset ($-1.88$ W m$^{-2}$) and J2014 mineralogy dataset ($-1.37$ W
m$^{-2}$) when using EMIT data, compared to the default dust mineralogy compositions. This
reduction could be due to more accurate mineralogical characterization, which alters the
dust's optical properties and reduces its ability to absorb and scatter sunlight. Conversely,
SWRF was enhanced in the atmosphere (N2012: $+1.44$ W m$^{-2}$, J2014: $+0.84$ W m$^{-2}$) when
using the EMIT data, indicating that the updated mineralogical information leads to a
different interaction with solar radiation in the atmospheric layer, possibly due to changes
in dust composition that affect scattering and absorption properties at higher altitudes.
Furthermore, SWRF at the TOA transitioned from negative to positive in simulations
using the EMIT data. For the N2012 dataset, the SWRF varied from $-1.73$ to $+1.59$ W m$^{-2}$,
and for the J2014 dataset, it ranged from $-2.14$ to $+0.07$ W m$^{-2}$. This shift suggests that
more accurate dust mineralogy data, particularly from satellite observations, can have a
significant impact on the amount of solar radiation reflected back to space, thereby
influencing the radiative balance at the TOA. The transition from negative to positive
forcing at the TOA emphasizes the importance of incorporating realistic mineralogical
information to enhance the accuracy of dust-related radiative forcing calculations and
better understand their role in climate systems.



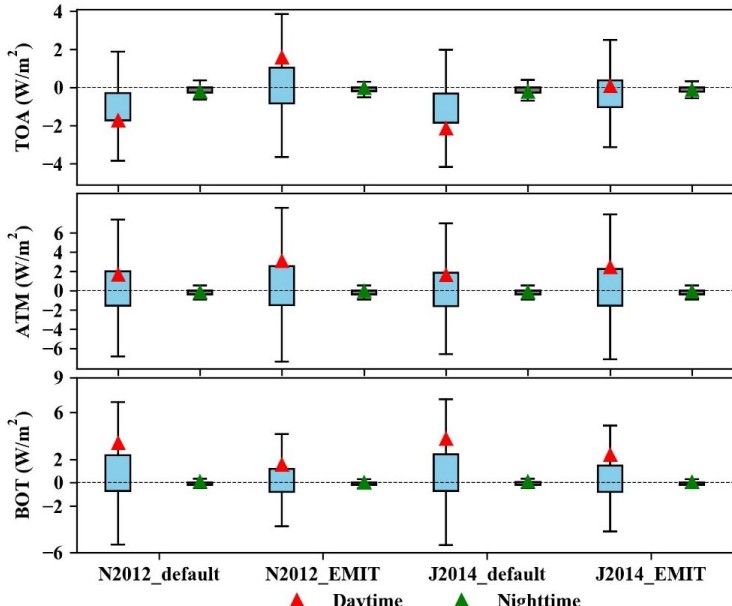


Figure 6. Day-night changes of TOA, ATM and BOT shortwave radiation forcings from simulations
using different composition atlases (N2012_default, N2012_EMIT, J2014_default and J2014_EMIT)
compared to bulk dust.

**3.3 Radiative effect on air quality**

Aerosol effects not only gave rise to changes in meteorological variables but also had

a significant impact on air quality. As shown in Figure 7, the inclusion of bulk dust aerosol

feedbacks in the WRF-CHIMERE model resulted in substantial increases in $PM_{10}$

concentrations, with an average increase of 119.48 μg m⁻³. This rise in particulate matter

highlights the important role of dust aerosols in contributing to local and regional air

pollution, especially in regions that are susceptible to dust storms. Along with these

increases in $PM_{10}$ concentrations, $O_3$ concentrations slightly decreased, with an average

reduction of −46.52 μg m⁻³. This reduction in ozone can be attributed to the complex

interaction between dust particles and ozone precursor gases, where dust aerosols can act

as both a sink for ozone and influence the photochemical processes that govern its

formation and degradation.

The most pronounced increases in $PM_{10}$ concentrations occurred in the Badain Jaran



Desert, a well-known dust source region, where peak values exceeded 1200 μg m⁻³. This
reflects the large dust emissions typical of desert environments, where strong winds
mobilize vast quantities of particulate matter. Downwind regions, including Ningxia,
Shaanxi, and Beijing, also experienced significant $PM_{10}$ elevations, with concentration
differences reaching approximately 600 μg m⁻³ compared to baseline levels. These
increases in $PM_{10}$ highlight the long-range transport of dust particles, which can impact air
quality far from the source regions and have implications for public health, especially in
urban areas.

While the use of speciated dust had some impact on long-range dust transport, its

overall effects on $PM_{10}$ were relatively limited. The inclusion of more detailed
mineralogical data allowed for a better representation of dust composition but did not
significantly alter the overall dust load in terms of $PM_{10}$ concentrations. This suggests that,
while dust speciation can influence the optical properties and radiative effects of dust, the
total mass concentration of dust particles in the atmosphere is primarily driven by factors
such as dust emission sources, atmospheric transport, and meteorological conditions, rather
than mineral composition alone.

Ozone changes along transport pathways were generally smaller than the surrounding

concentrations, typically ranging from −60 to −40 μg m⁻³. These smaller changes in $O_3$
concentrations reflect the fact that dust aerosols have a more localized and complex effect
on ozone formation and destruction, with significant variability depending on the regional
and temporal context. In particular, dust-induced reductions in ozone are likely to be
influenced by the local presence of other atmospheric constituents, such as nitrogen oxides
and volatile organic compounds, which play a key role in ozone chemistry.

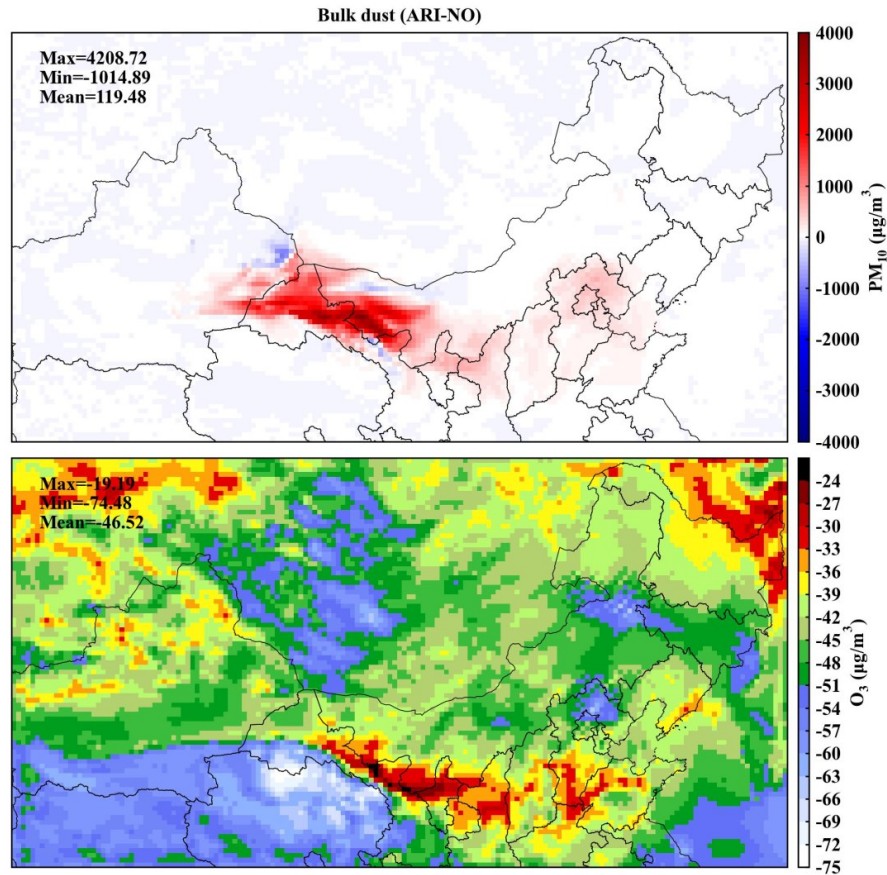


Figure 7. Changes in PM$_{10}$ and O$_3$ concentrations resulting from bulk dust-induced ARI effects, compared to the scenario without aerosol feedbacks.


The spatial differences in PM$_{10}$ and O$_3$ concentrations simulated by WRF-CHIMERE
with different mineralogy atlases compared to bulk dust, enabling ARI effects, are depicted
in Figures 8 and A4. These comparisons reveal substantial changes in the PM$_{10}$ and O$_3$
concentrations across the different mineralogical compositions, including N2012_Default,
N2012_EMIT, J2014_Default, J2014_EMIT, and bulk dust. This suggests that the
normalization of the 12 minerals from these atlases significantly modifies meteorological
conditions, further influencing the relative abundances of dust particles and their
subsequent effects on air quality and atmospheric composition.

balanced





When compared to bulk dust, reduced $PM_{10}$ concentrations were primarily observed in the Taklimakan Desert, with decreases of around 60 μg m⁻³, while increases in $PM_{10}$ concentrations occurred in the Badain Jaran Desert and its downwind regions, with concentrations rising up to 1000 μg m⁻³. These regional variations indicate that different dust mineralogical compositions can impact the emission and transport of dust, with certain mineral types leading to more efficient scattering or absorption of radiation, which may alter the local meteorological conditions and dust dispersion patterns.

For $O_3$ concentrations, reductions and enhancements were mainly observed in the Horqin sandy land and North China Plain, with changes up to 4 μg m⁻³, respectively. This highlights the complex interaction between dust aerosols and ozone chemistry, where dust can either enhance or reduce ozone concentrations depending on the region. Dust aerosols can influence ozone levels by acting as a surface for heterogeneous chemical reactions or by modifying the photochemical processes that control ozone formation and destruction.

When considering the EMIT data, $PM_{10}$ concentrations were reduced in dust source regions and enhanced in downwind regions, with reductions of up to −1567.16 μg m⁻³ and increases of +218.26 μg m⁻³. This suggests that more accurate mineralogical data can influence dust transport patterns, leading to greater reductions in $PM_{10}$ at the source regions and increased dust concentrations in the downwind areas. These findings further emphasize the role of mineralogical composition in modulating dust aerosol behavior and distribution.

For $O_3$, enhancements appeared in source regions, while reductions were observed in downwind regions, with changes ranging from −2.46 to +3.52 μg m⁻³. These trends suggest that more accurate dust speciation can influence regional ozone levels in different ways, with possible implications for local air quality and atmospheric chemistry. Notably, the impacts on $PM_{10}$ concentrations from N2012_EMIT compared to N2012_Default were larger than those observed from J2014_EMIT versus J2014_Default, while the impacts on $O_3$ concentrations followed the opposite trends. This indicates that the choice of dust mineralogical dataset has a differential impact on $PM_{10}$ and $O_3$ concentrations,





underscoring the importance of considering mineral composition in aerosol modeling to
more accurately predict air quality and climate effects.


Figure 8. Difference in PM$_{10}$ concentrations considering bulk dust and various dust mineralogy atlases
that enable ARI effects.

Figure 9 shows the percentage changes in surface concentrations of mineral dust with
and without considering ARI effects. These results provide valuable insight into how the
inclusion of ARI effects modifies the composition and radiative properties of dust aerosols,
depending on the mineralogical dataset used. For the N2012_default and N2012_EMIT
data, quartz and feldspar accounted for a substantial portion of the total dust, ranging from
approximately 51.7% to 57.4% for quartz and 18.6% to 19.8% for feldspar. This indicates
that quartz and feldspar are the dominant mineral components in the dust modeled with the
N2012 dataset.
In contrast, for the J2014_default dataset, the mineral composition was more
diversified, with calcite, quartz, and mica contributing about 26.3%, 24.0%, and 20.0%,
respectively, to the total dust composition. This shift in mineral proportions reflects the
differences in the mineralogical characterization between the N2012 and J2014 datasets,
with J2014 incorporating a broader range of dust minerals. For J2014_EMIT, the mineral



composition shifted further, with quartz and mica making up approximately 46.8% and
27.5% of the dust, respectively. This highlights the importance of using accurate
mineralogical data, such as that from EMIT satellite observations, to better represent the
composition of dust aerosols in simulations.

The inclusion of EMIT data led to an increase in the absorption percentage of hematite

by about 8% for N2012 and 6% for J2014. Hematite is a highly absorbing mineral,
especially in the shortwave spectrum, and its increased presence enhances the dust's ability
to absorb solar radiation, thereby affecting the DRE in the shortwave spectrum. This
increase in hematite absorption is significant, as it directly impacts the radiative effects of
dust, potentially contributing to a greater cooling effect on the atmosphere by modifying
the shortwave radiation balance.

While quartz constitutes the largest portion of the dust in both the N2012 and J2014

datasets, its DRF effects are relatively limited, as noted in Li et al. (2021). Quartz is known
for its high reflectivity in the shortwave spectrum, and while it makes up a large fraction
of the total dust mass, it has a less pronounced effect on radiative forcing compared to more
absorbing minerals like hematite or mica. This suggests that, despite its dominance in dust
composition, quartz plays a smaller role in modifying the energy balance of the atmosphere
through direct radiative effects.




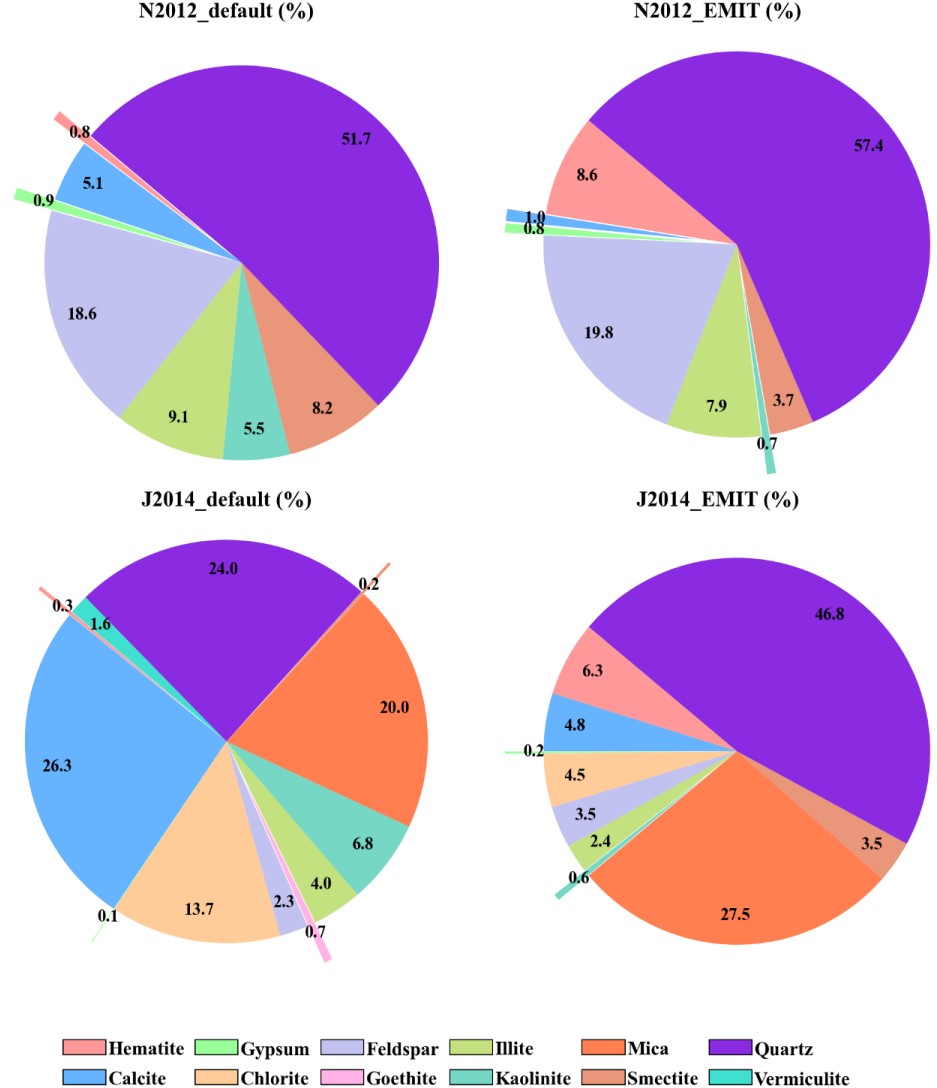

Figure 9. Contributions of different mineralogical compositions using N2012_default, N2012_EMIT, J2014_default, and J2014_EMIT, considering ARI effects, compared to the scenario without enabling aerosol feedbacks.

**3.4 Limitations and uncertainties for aerosol feedbacks of mineralogical dust**

The accuracy of simulated dust emission is intricately linked to soil properties, such as soil texture and moisture, which primarily influence the threshold friction velocity required for dust particle mobilization (Kim and Choi, 2015; Su and Fung, 2015). These



factors play a critical role in determining the magnitude and spatial distribution of dust
emissions, underscoring the need for precise and high-resolution soil data in dust modeling.
While the current EMIT L3 data offers a spatial resolution of 0.5° × 0.5°, real-time higher
spatial resolution datasets, such as the 60 m × 60 m EMIT L2B mineral atlas
(https://earth.jpl.nasa.gov/emit/internal_resources/282), can provide a more detailed
representation of soil mineralogy, thereby enhancing the fidelity of dust emission
simulations.

The uncertainty associated with dust mineralogical datasets is being actively

addressed by NASA's EMIT. This initiative has deployed a hyperspectral imaging
spectrometer aboard the International Space Station to deliver global retrievals of soil
mineral compositions with unprecedented spatial detail. The spectrometer captures spectral
absorption features within the UV to near-infrared range (0.38–2.5 μm), offering critical
insights into the distribution and variability of soil minerals (Castellanos et al., 2024;
Connelly et al., 2021). In addition to satellite-based observations, ground-based stations
play a vital role in measuring dust mineralogical compositions using stationary instruments,
which provide localized and highly accurate data. Complementing these measurements,
aircraft-based instruments offer the capability to sample dust particles along specific flight
tracks, providing valuable vertical and spatial profiles of speciated dust properties (e.g.,
size and mixing state, Panta et al., 2023; Ryder et al., 2015). Together, these observational
platforms form a robust foundation for validating and improving dust models.

Aerosol-cloud interactions involving speciated dust are another critical aspect of dust-

climate interactions that require further investigation, especially for feldspar and quartz
(Atkinson et al., 2013; Chatziparaschos et al., 2023). Incorporating these interactions into
two-way coupled WRF-CHIMERE models can provide a more comprehensive
understanding of the feedback mechanisms between dust aerosols and cloud microphysics.
Such implementations are currently a focus of ongoing work, aiming to refine the
representation of dust-induced radiative and microphysical effects in regional and global





models. These efforts will not only improve model accuracy but also enhance our ability
to predict the impacts of dust on weather, air quality, and climate.

**4 Conclusion**
This study has provided a comprehensive analysis of the role of dust mineral
composition in atmospheric radiation and pollution in North China, using a two-way
coupled WRF-CHIMERE model integrated with three mineralogical datasets (N2012,
J2014, and L2024 from EMIT). The research focused on the March 2021 dust storm event
to evaluate the ARI effects and their impacts on regional meteorology and air quality.
The findings revealed significant spatial variations in radiative forcing due to
differences in dust mineralogy. Compared to the ARI effects of bulk dust, the mineralogical
composition of dust aerosols can increase SW radiation forcing at the surface and in the
atmosphere by +0.10 to +0.82 W m$^{-2}$, while simultaneously causing a decrease of
approximately −0.72 W m$^{-2}$ in SW radiation forcing at the TOA. Integrating EMIT data
into the model reduced PM$_{10}$ biases by over 15% in high-concentration regions and
improved ozone predictions, with localized changes ranging from −2.46 to +3.52 µg m$^{-3}$.
Specifically, the ARI effects of these mineralogical compositions led to a notable increase
in PM$_{10}$ levels, reaching up to 1189.48 µg m$^{-3}$ in dust source regions, when compared to
bulk dust scenarios.
These findings highlight the critical importance of considering mineralogical data in
improving simulations of dust-related radiative forcing and air quality impacts. High-
resolution observational data, such as EMIT satellite observations, combined with
sensitivity studies that account for a wider range of observational factors, including
atmospheric conditions under varying aerosol optical depth and water vapor loading, as
well as the spectral representation of surface mineralogical features, along with alternative
parameterizations of instrument noise of variable signal-to-noise and spectral sampling or
entirely different mineral identification algorithms, are crucial for improving atmospheric
models for dust simulations. Additionally, this study emphasizes the need for a more



nuanced understanding of the feedback mechanisms between dust mineral composition and
cloud microphysics, which can significantly influence regional climate dynamics and air
quality.
Despite the robust methodology and comprehensive analysis, several limitations are
acknowledged. The model exhibited systematic biases in simulating surface radiation
fluxes, near-surface wind dynamics, and temperature fields, particularly in North China.
These biases suggest challenges in accurately capturing the complex interactions between
dust aerosols and the atmosphere. The choice of mineralogical datasets and modeling
framework also significantly affected the calculated dust radiative forcings, indicating a
need for further refinement and validation of these datasets.
Future research should focus on integrating aerosol-cloud interactions of speciated
dust into two-way coupled models to enhance our understanding of feedback mechanisms
between dust aerosols and cloud microphysics. Furthermore, the development and
incorporation of higher-resolution soil data and real-time satellite observations could
further refine dust emission simulations and reduce model biases.

**Data and software availability**
The meteorological ICs and BCs, Chemical ICs and BCs and emission data used for
WRF–CHIMERE and all data used to create figures and tables in this study are provided
in an open repository on Zenodo (https://doi.org/10.5281/zenodo.14728874, Gao et al.,
2025a). Himawari and CALIPSO satellite data are available at
ftp://ftp.ptree.jaxa.jp/jma/netcdf and https://subset.larc.nasa.gov/calipso, respectively.
The source codes of the two-way coupled WRFv3.7.1–CHIMERE v2020r3 models
are obtained from https://www.lmd.polytechnique.fr/chimere. The related source codes,
configuration information, namelist files and automated run scripts of these three two-way
coupled models are archived at Zenodo with the following associated DOI:
https://doi.org/10.5281/zenodo.14729124 (Gao et al., 2025b).



**Author contributions**

CG, XZ, HY and LH carried out the data collection, related analysis, figure plotting, and paper writing. HZ, SZ, and AX were involved with the original research plan and made suggestions for the paper writing.

**Competing interests**

The contact author has declared that neither they nor their co-authors have any competing interests.

**Acknowledgments**

This study was financially sponsored by the National Natural Science Foundation of China (grant nos. 42305171, 42371154 & 42171142), the Natural Science Foundation of Jilin Province (YDZJ202201ZYTS476), the National Key Scientific and Technological Infrastructure project "Earth System Numerical Simulation Facility" (2023-EL-PT-000469), the Youth Innovation Promotion Association of Chinese Academy of Sciences, China (grant nos. 2022230), the National Key Research and Development Program of China (grant nos. 2017YFC0212304 & 2019YFE0194500) and the Talent Program of Chinese Academy of Sciences (Y8H1021001).

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

**Appendix**
Table A1. Model setups and inputs for the WRF-CHIMERE model.

|  |  | WRF-CHIMERE |
| --- | --- | --- |
| Domain configuration | Horizontal grid spacing | 27 km (165 × 87) |
|  | Vertical resolution | 33 levels |
| Physics parameterization | Shortwave radiation | RRTMG |
|  | Longwave radiation | RRTMG |
|  | Cloud microphysics | Thompson |
|  | PBL | YSU |
|  | Cumulus | Grell-Freitas |
|  | Surface | Monin-Obukhov |
|  | Land surface | Noah LSM |
|  | Icloud | Xu-Randall method |
| Chemistry scheme | Aerosol mechanism | SAM |
|  | Aerosol size distribution | Sectional (10 bins) |
|  | Aerosol mixing state | Core-Shell |
|  | Gas-phase chemistry | MELCHIOR2 |
|  | Photolysis | Fast-JX with cloud effects |
| Emission | Dust emission | Kok |
| Input data | Meteorological ICs and BCs | FNL |
|  | Chemical ICs and BCs | LMDZ-INCA |





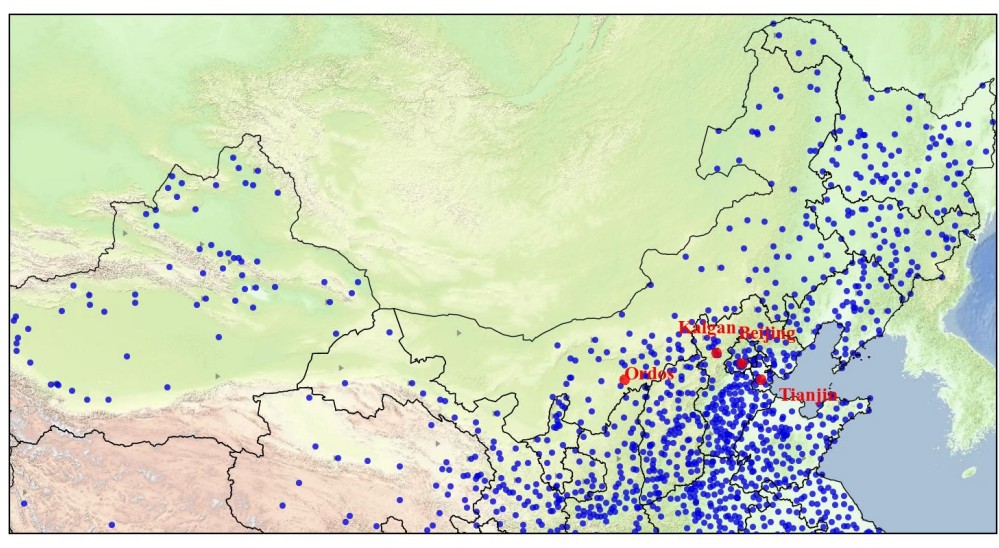

- ▶ **Solar radiation monitoring station**   ● **Air quality monitoring station**
- ● **Meteorological monitoring station**


Figure A1. Simulation domain and locations of meteorology and air quality monitoring stations.




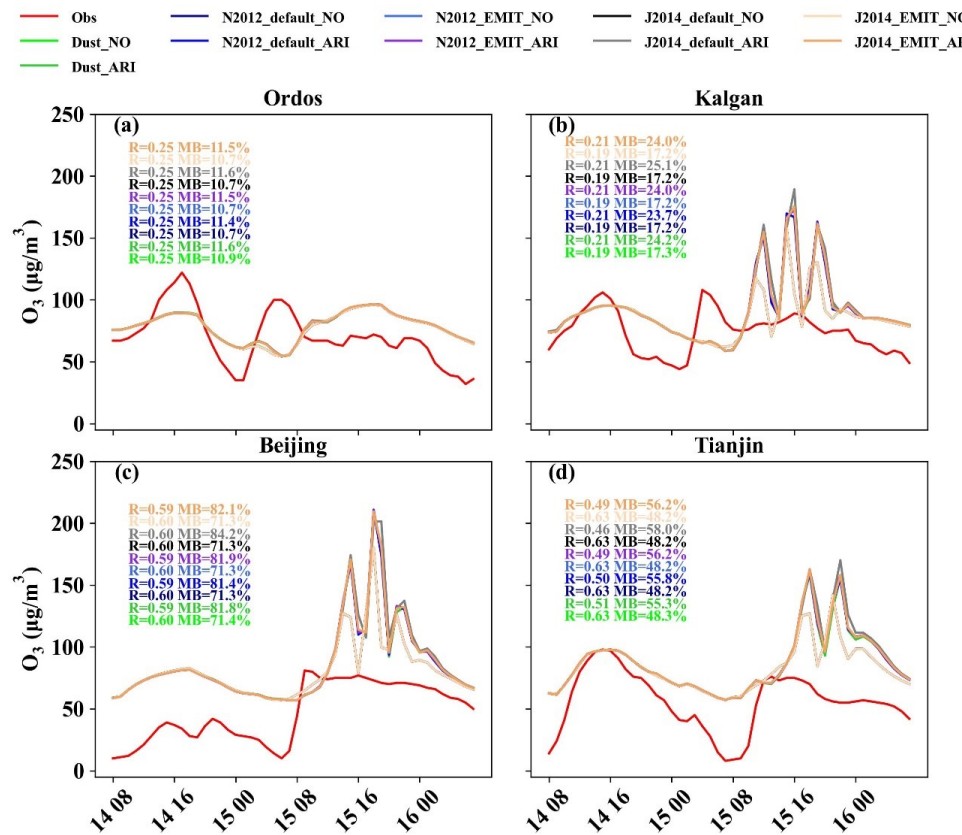

Figure A2. Statatiscal metrices between observed and simulated O₃ concentrations by different
scenario simulations.



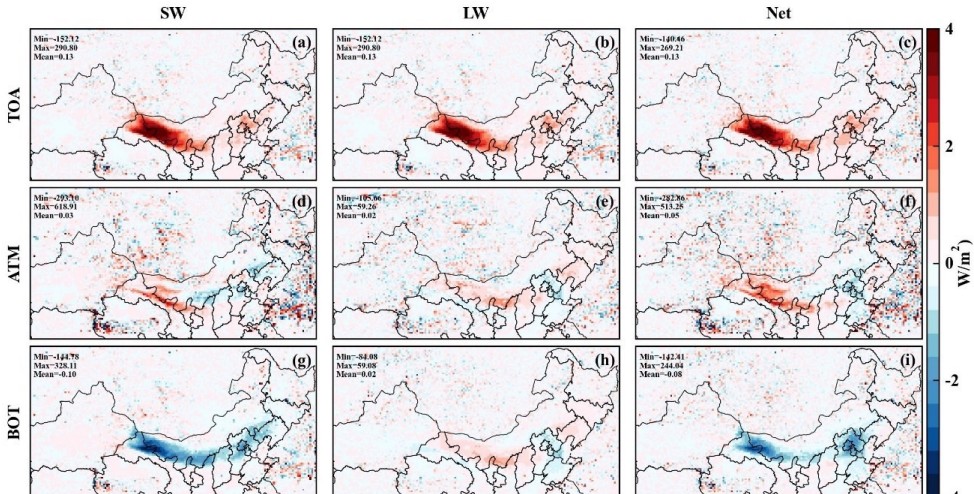

Figure A3. Difference between TOA, ATM and BOT radiation forcings with considering J2014 and N2012 mineralogical dust compositions (i.e., N2012_default) enabling ARI effects.

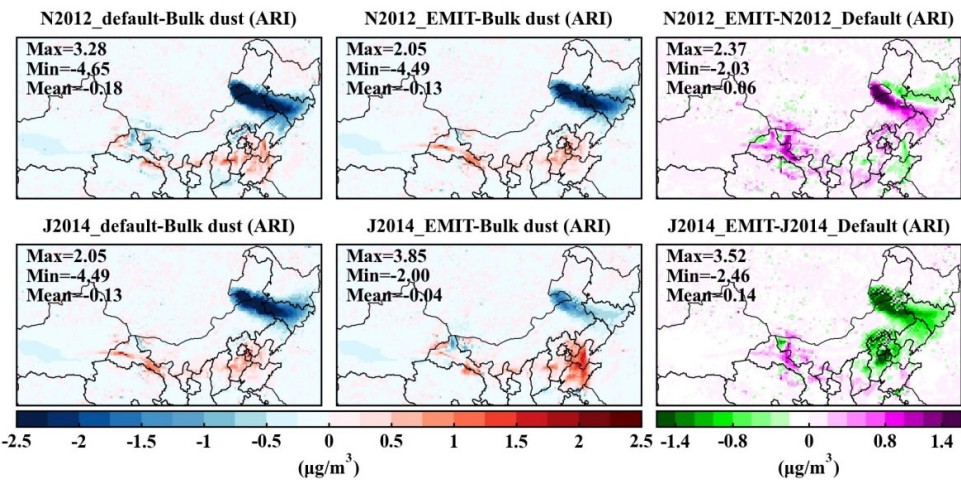

Figure A4. Difference in O₃ concentrations considering bulk dust and various dust mineralogy atlases that enable ARI effects.



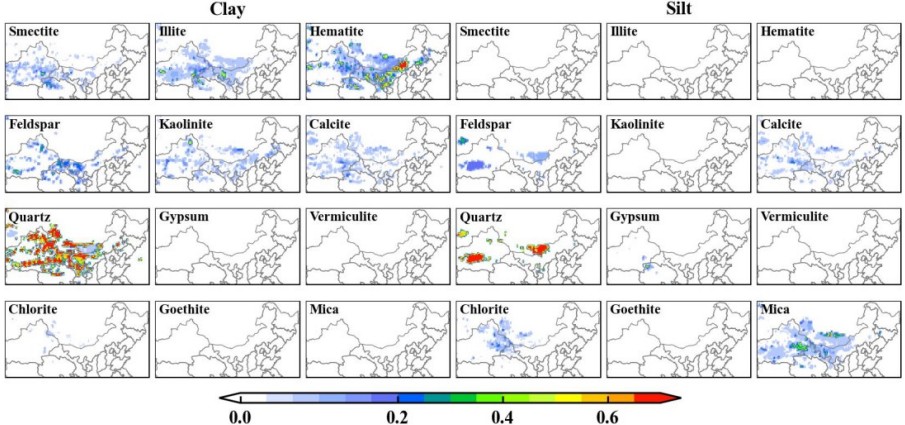

Figure A5. Spatial distribution of content for the different mineral dust species in the silt and clay fraction of the soil for J2014 with EMIT satellite data.

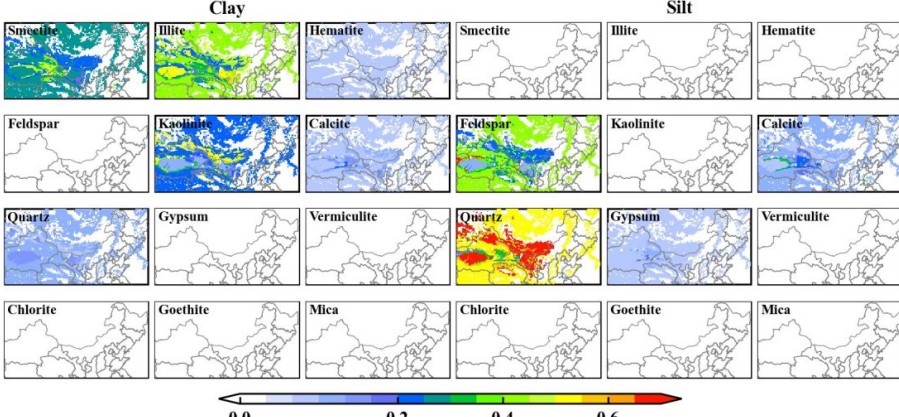

Figure A6. Spatial distribution of content for the different mineral dust species in the silt and clay fraction of the soil for original N2012 mineralogical data.



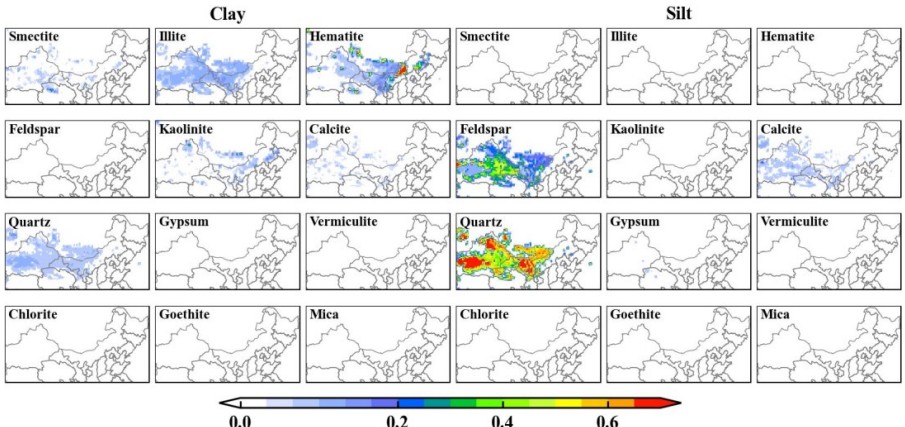

Figure A7. Spatial distribution of content for the different mineral dust species in the silt and clay fraction of the soil for N2012 with EMIT satellite data.