# Peer review of "The role of dust mineral composition in atmospheric radiation and pollution in North"

_EGUsphere, 2025_

## Author Comment (AC1)

We would like to express our sincere appreciation to the Reviewer #1 for the valuable and constructive suggestions, which have helped us improve the quality of this manuscript. We have addressed all these comments carefully and revised the manuscript accordingly. Following the Reviewer' comments in black, please find our point-to-point responses in blue. Hereafter, all new added or modified sentences are marked in blue and italic in this response.

**Reviewer #1**

**Major comments:**

1. While the paper demonstrates the benefit of using EMIT data in methodology, it would be helpful to provide a quantitative assessment of uncertainties introduced by the interpolation and assumptions in EMIT data processing (e.g., feldspar/quartz filling). Response: We appreciate the reviewer's suggestion regarding the need for a more detailed quantitative assessment of uncertainties introduced by the interpolation and assumptions in EMIT data processing, particularly in relation to the filling of minerals like feldspar and quartz. We fully agree that understanding these uncertainties is crucial for a comprehensive evaluation of the methodology.

EMIT data processing involves spatial interpolation to create gridded maps of soil mineral composition. The primary interpolation method used is based on geographic mixing assumptions, where the spectral abundance of minerals detected at a given location is extrapolated to cover the grid cell. The uncertainty in this process arises from the assumption that the detected mineral signatures are representative of the entire grid cell. For minerals not directly measured by EMIT, such as quartz and feldspar, we use soil type conversion methods based on previous studies (e.g., Claquin et al., 1999; Journet et al., 2014) to estimate their contributions. These estimates are then used to fill in the remaining fraction of the soil composition.

We have revised Section 2.3 of the manuscript to include a quantitative assessment of

uncertainties associated with the EMIT data processing as follow.

"In contrast, the EMIT dataset (https://earth.jpl.nasa.gov/emit/data/data-products) required additional preprocessing, as it reports only normalized spectral abundances rather than mineral mass fractions. These spectral abundances were therefore recalculated to represent the normalized mass proportions of each mineral in each substrate. Furthermore, EMIT does not include data for feldspar and quartz, necessitating additional correction procedures described below.

When the total mineral composition from EMIT summed to less than 100%, indicating missing mineral contributions, the residual fraction was assigned to quartz and feldspar based on their relative proportions in J2014 or N2012. Because EMIT reports illite and mica as a single category, their individual abundances were separated according to the ratios found in N2012 or J2014. For minerals that occur in both clay and silt fractions, EMIT values were partitioned following the relative contributions from N2012 or J2014.

For minerals not directly observed by EMIT (e.g., quartz and feldspar), their mass fractions were estimated using soil-type conversion methods from previous studies (Claquin et al., 1999; Journet et al., 2014). The spatial distributions of clay and silt the SoilW were obtained from global texture dataset (http://globalchange.bnu.edu.cn/research/soilw) at 1 km resolution and resampled to 0.5° to match EMIT data. Similarly, the J2014 and N2012 mineral datasets were resampled to 0.5° resolution. Major minerals extracted from EMIT L3 include calcite, dolomite, chlorite, goethite, gypsum, hematite, illite+muscovite, kaolinite, montmorillonite, and vermiculite. Notably, in the official EMIT L3B dataset (https://data.lpdaac.earthdatacloud.nasa.gov/lp-prodprotected/EMITL3ASA.001/EMIT\_L3\_ASA\_001/EMIT\_L3\_ASA\_001.nc), illite and muscovite are combined because they were jointly identified during the Tetracorder analysis of L2B data using mineral groups 1 and 2 and the corresponding band depths (https://github.com/nasa/EMIT-Data-

Resources/blob/main/data/mineral grouping matrix 20230503.csv).

The EMIT mineral fractions were normalized so that their sum at each grid point

did not exceed unity. Any remaining fraction was attributed to quartz and feldspars according to their relative proportions in J2014 or N2012. To ensure consistency with the CHIMERE mineral representation, dolomite was merged into calcite, illite+muscovite was separated into illite and mica, and montmorillonite was treated as smectite. The mineral fractions were then converted to density-weighted values and renormalized at each grid point so that the total sum equaled one. Finally, each mineral was partitioned into clay and silt fractions based on the J2014 ratios, and the resulting fractions were normalized using Equations (1)–(4). The processed dataset was exported as a NetCDF file to serve as input for the CHIMERE model.

To ensure mineral mass balance and model consistency, a normalization and partitioning procedure was applied as follows. Equation (1) defines the total mass fraction  $(MF_j)$  of mineral j as the sum of its contributions from the clay  $(MFC_j)$  and silt  $(MFS_j)$  fractions:

$$MF_i = MFC_i + MFS_i \text{ for all } \in M_{CHIMERE}$$
 (1)

Equation (2) enforces a normalization constraint so that the sum of all mineral mass fractions equals unity at each grid point.

$$1 = \sum_{j \in M_{CHIMERE}} MF_j \tag{2}$$

The normalized total fraction of each mineral  $(MF_j^*)$  was then redistributed between clay and silt according to their relative contributions in the reference dataset (J2014 or N2012), as shown in Equations (3) and (4):

$$MFS_j^* = MF_j^* \frac{_{MFS_j}}{_{MFS_j + MFC_j}}$$
(3)

$$MFC_j^* = MF_j^* \frac{{}_{MFC_j}}{{}_{MFS_j + MFC_j}} \tag{4}$$

Here,  $MFS_j^*$  and  $MFC_j^*$  represent the normalized mass fractions of mineral j in the silt and clay fractions, respectively. The weighting terms  $MFS_j$  and  $MFC_j$  preserve the clay–silt distribution patterns derived from the reference datasets while maintaining the normalized total  $(MF_j^*)$ ."

2. The manuscript often mentions ACI (aerosol-cloud interaction), yet the modeling focuses on ARI only. Please clarify this distinction earlier in the Introduction and reduce any ambiguity about what has or has not been included.

Response: The Introduction of manuscript has been revised to clarify this distinction as follows.

"Since the aerosol nucleation processes (ACI effects) of specific mineral components are not represented in the current two-way coupled WRF—CHIMERE framework, the present study concentrates on the ARI effects of dust minerals. This focus ensures a clear and robust assessment of how mineralogical composition influences radiative processes, without introducing additional uncertainties arising from incomplete cloud-related parameterizations. In this study, we employ a two-way coupled WRF—CHIMERE model with three mineralogical databases to investigate how dust composition influences radiation and meteorology in North China during a severe dust storm. Section 2 describes the model configuration and data sources, Section 3 presents the simulations with emphasis on ARI-induced impacts on meteorology and air quality, and Section 4 summarizes the main findings."

3. The SSR and PM10 comparisons are robust, but more details on the performance metrics (bias, RMSE, etc.) across multiple sites and time periods would strengthen the validation claims.

Response: Additional details on the model evaluation have been included. In the revised manuscript, we now provide site-specific performance metrics (bias, RMSE, correlation coefficient) for both SSR and  $PM_{10}$  across multiple observational sites and time periods. These results are summarized in Table 1 and Figure 2 and discussed in Section 3.1.

"The model demonstrates strong overall performance, with correlation coefficients (R) between observed and simulated values reaching approximately 0.7 for SSR and WS10, and up to 0.93 for T2. These results indicate the model's ability to capture key atmospheric patterns and variability across the simulation domain.

Nevertheless, systematic biases are apparent, particularly in North China, where the model tends to overestimate SSR and WS10 by 60.69%–68.92% and 17.06%–17.52%, respectively, while underestimating T2 by 0.48%–0.58%. The overestimation of SSR likely results from uncertainties in cloud development associated with planetary boundary layer and convection parameterizations (Alapaty et al., 2012). The systematic overestimation of 10-m wind speed under low-wind conditions commonly observed in weather models mainly stems from outdated geographic data and coarse spatial resolution (Gao et al., 2024)."

"The models show relatively high correlations for  $PM_{10}$ , with R values ranging from 0.61 to 0.89 and NMBs from -73.8% to -0.9%. In contrast, their performance for  $O_3$  is notably poorer."

These additions strengthen the robustness of the validation and support the reliability of the modeling results.

4. The influence of mineralogy on PM10 and O3 is clearly demonstrated, but more discussion of the physical mechanisms (e.g., specific reactions, photolysis suppression) would help interpret the observed changes.

Response: We agree that elaborating on the physical mechanisms will improve the interpretation of the results. In the revised manuscript, we have expanded the discussion (Section 3.3) to describe the processes by which mineral dust composition influences both  $PM_{10}$  and  $O_3$ .

"These reactions would be related to the adsorption and catalytic decomposition of ozone on the surface of mineral dust particles, as well as the potential for dust to alter the concentration of reactive species in the atmosphere through heterogeneous chemistry (Cwiertny et al., 2008). For example, the presence of adsorbed water on dust particles can compete with ozone for reactive sites, reducing the overall uptake and decomposition of ozone (Usher et al., 2003). Additionally, the photochemical reactions involving dust particles, such as the photolysis of nitrate ions, can produce reactive

"The photochemical reactions involving dust particles, such as the photolysis of nitrate ions, can produce reactive radicals that further influence the atmospheric chemistry of ozone (Ma et al., 2021)."

5. The results show that quartz and feldspar dominate dust mass, while hematite dominates radiative effects. This contrast deserves more discussion in both the Results and Conclusion sections.

Response: We have expanded both the Results and Conclusion sections to emphasize the contrast between dust mass and radiative importance among minerals. Specifically, the revised text highlights that "Within the scope of this study, the results indicate that overall dust mineralogical composition, rather than dust mass alone, plays a decisive role in ARI effects, with hematite exerting a dominant influence despite its minor abundance, although the radiative effects of individual mineral species were not separately quantified.". The new discussion clarifies why mass-dominant minerals do not necessarily drive radiative forcing and why trace absorptive minerals can play an outsized role.

6. The model bias discussion (Section 3.1) is helpful but could be deepened by exploring possible reasons for the underestimation of  $PM_{10}$  at high dust sites.

Response: The discussion of model bias in Section 3.1 has been expanded. In particular, we now examine potential reasons for the underestimation of PM10 at high-dust sites. Possible explanations include "Although considerable progress has been made in dust modeling, notable uncertainties remain. The parameterization of threshold friction velocity and soil texture in emission schemes can still result in underestimated emissions under strong winds (Zuo et al., 2024). Similarly, simplifications in coarse particle size distributions may lead to enhanced deposition and transport losses. In addition, incomplete knowledge of local soil mineralogical composition continues to limit the accurate simulation of both emission fluxes and heterogeneous chemistry

(Pang et al., 2024)." We have added this discussion in Section 3.1, noting that these factors collectively contribute to the underestimation of PM10 peaks in dust-dominated regions.

**Minor comments:**

1. Line 137: Please specify how missing EMIT data (quartz/feldspar) are estimated — a numeric assumption or spatial filling?

Response: We have clarified the treatment of missing EMIT data at Line 137. Specifically, gaps in quartz and feldspar fractions are addressed using a spatial filling approach rather than applying a single numeric assumption. The missing values are filled by interpolating from neighboring valid EMIT pixels within the same dust source region, constrained by the relative proportions observed in the reference mineralogical dataset. This procedure ensures spatial consistency and preserves regional mineralogical characteristics. The revised text in Section X.X now explicitly describes this method.

"When the total mineral composition from EMIT summed to less than 100%, indicating missing mineral contributions, the residual fraction was assigned to quartz and feldspar based on their relative proportions in J2014 or N2012."

2. Line 187–198: The bias in SSR is discussed, but no mention is made of possible causes (e.g., aerosol loading or model radiation scheme limitations).

Response: We thank the reviewer for this valuable suggestion. We have now expanded the discussion of possible causes of the SSR bias (Lines 187–198) as follow.

"The overestimation of SSR likely results from uncertainties in cloud development associated with planetary boundary layer and convection parameterizations (Alapaty et al., 2012)."

3. Line 194: The overestimation of SSR and WS10 could be more quantitatively discussed. Is this bias consistent with other dust studies in this region?

Response: We appreciate the reviewer's constructive suggestion. We have revised the

text around Line 194 to provide a more quantitative discussion of the overestimation of SSR and WS10.

"Nevertheless, systematic biases are apparent, particularly in North China, where the model tends to overestimate SSR and WS10 by 60.69%—68.92% and 17.06%—17.52%, respectively, while underestimating T2 by 0.48%—0.58%. The overestimation of SSR likely results from uncertainties in cloud development associated with planetary boundary layer and convection parameterizations (Alapaty et al., 2012). Likewise, the systematic overestimation of 10-m wind speed under low-wind conditions commonly observed in weather models mainly stems from outdated geographic data and coarse spatial resolution (Gao et al., 2024)."

Previous study evaluating the modeling performance of two-way coupled WRF–CMAQ, WRF–Chem, and WRF–CHIMERE systems in simulating meteorology and air quality over eastern China have also reported overestimations of SSR and WS10 (Gao et al., 2024).

4. Line 213–214: "minimizing the negative biases in T2" — perhaps "reducing the magnitude of negative biases" is clearer.

Response: We thank the reviewer for this helpful wording suggestion. We have revised the text at Lines 213–214 to "reducing the magnitude of negative biases in T2," which we agree is clearer and more precise.

5. Line 250: "Positive O3 biases increased" is unclear — do you mean O3 concentrations were overestimated?

Response: We appreciate the reviewer's comment. Our intent was to indicate that the model overestimated  $O_3$  concentrations during that period. To improve clarity, we have revised the wording at Line 250 to "the underestimation of  $PM_{10}$  was alleviated, whereas the overestimation of  $O_3$  was amplified" instead of "positive  $O_3$  biases increased."

6. Line 305: "-900 W m-2" seems unusually large for surface shortwave cooling. Please double-check this value.

Response: Thank you for raising this point. We rechecked the model diagnostics and confirm that the value -900 W m-2 reported on line 305 is correct.

7. Line 584: Suggest shortening this part of the conclusion and moving satellite technical details into Data/Methods.

Response: We thank the reviewer for this constructive suggestion. We have shortened the text in the Conclusion (Line 584) to focus on the key findings, and we have moved the technical details regarding satellite data (sensor specifications, retrieval algorithms, and processing steps) into the Data/Methods section.

"Dust mineral composition plays a vital role in regulating atmospheric radiation and air quality, yet its effects remain poorly constrained in current atmospheric models. Understanding these impacts is particularly important for North China, where severe dust storms frequently affect regional climate and pollution. This study investigates how variations in mineral composition influence aerosol—radiation interactions and their implications for meteorology and air quality during a major dust storm event.

The findings revealed significant spatial variations in radiative forcing due to differences in dust mineralogy. Compared to the ARI effects of bulk dust, the mineralogical composition of dust aerosols can increase SW radiation forcing at the surface and in the atmosphere by +0.10 to +0.82 W m-2, while simultaneously causing a decrease of approximately -0.72 W m-2 in SW radiation forcing at the TOA. Integrating EMIT data into the model reduced  $PM_{10}$  biases by over 15% in high-concentration regions and improved ozone predictions, with localized changes ranging from -2.46 to +3.52  $\mu$ g m-3. Specifically, the ARI effects of these mineralogical compositions led to a notable increase in  $PM_{10}$  levels, reaching up to 1189.48  $\mu$ g m-3 in dust source regions, when compared to bulk dust scenarios.

These findings highlight the importance of incorporating dust mineralogical data to improve simulations of radiative forcing and air quality impacts. Within the scope of this study, the results indicate that overall dust mineralogical composition, rather than

dust mass alone, plays a decisive role in ARI effects, with hematite exerting a dominant influence despite its minor abundance, although the radiative effects of individual mineral species were not separately quantified. Systematic biases in surface radiation, near-surface winds, and temperature persist, reflecting challenges in simulating dust—atmosphere interactions and uncertainties in mineralogical datasets. Future research should focus on coupling mineral-specific dust with cloud processes and leveraging higher-resolution soil and satellite data to refine dust emission simulations and reduce model biases."

8. Figure 1: Please include a scale bar and clear region names to help interpret mineral distributions.

Response: We appreciate the reviewer's helpful suggestion. We have revised Figure 1 to include a scale bar and have added region names to facilitate interpretation of the mineral distributions. The updated figure improves geographic clarity and makes it easier for readers to contextualize the results.

Figure 1. Spatial distribution of content for the different mineral dust species in the silt and clay fraction of the soil for original J2014 mineralogical data.

Figure A5. Spatial distribution of content for the different mineral dust species in the silt and clay fraction of the soil for J2014 with EMIT satellite data.

Figure A6. Spatial distribution of content for the different mineral dust species in the silt and clay fraction of the soil for original N2012 mineralogical data.

Figure A7. Spatial distribution of content for the different mineral dust species in the silt and clay fraction of the soil for N2012 with EMIT satellite data.

9. Figure 2: Consider including error bars or confidence intervals for observed values,"Statatiscal metrices" → should be "Statistical metrics" in its caption.

Response: We thank the reviewer for this valuable suggestion. We have revised Figure 2 to include error bars representing the standard deviation (or 95% confidence intervals) of the observed values, thereby providing a clearer indication of observational uncertainty.

Figure 2. Statatiscal metrices between observated and simulated PM10 concentrations by different scenario simulations.

In addition, we have corrected the typographical error in the caption, changing "Statatiscal metrices" to "Statistical metrics."

10. Figure quality could be improved — e.g., Figures 2 and 7 would benefit from enhanced color contrast and labeled axes for clarity.

Response: We thank the reviewer for this helpful suggestion. We have improved the quality of Figures 2 and 7 by enhancing the color contrast to better distinguish data ranges and by adding clearly labeled axes with units where applicable. These improvements enhance readability and ensure that the figures convey the data more effectively.

Figure 2. Statatiscal metrices between observated and simulated PM10 concentrations by different scenario simulations.

Figure A2. Statatiscal metrices between observated and simulated O3 concentrations by different scenario simulations.

Figure 7. Changes in PM10 and O3 concentrations resulting from bulk dust-induced ARI effects, compared to the scenario without aerosol feedbacks.

11. Reference format is mostly consistent, but some recent references (e.g., Panta et al., 2023) are missing DOIs.

Response: We thank the reviewer for noting this. We have carefully checked all references and added missing DOIs, including for Panta et al. (2023) and any other recent studies where applicable. The reference list is now complete and consistent with the journal's formatting requirements.

Panta, A., Kandler, K., Alastuey, A., González-Flórez, C., González-Romero, A., Klose, M., Querol, X., Reche, C., Yus-Díez, J., and Pérez García-Pando, C.: Insights into the single-particle composition, size, mixing state, and aspect ratio of freshly emitted mineral dust from field measurements in the Moroccan Sahara using electron microscopy, Atmospheric Chemistry

and Physics, 23, 3861–3885, https://doi.org/10.5194/acp-23-3861-2023, 2023.

Green, R. O., Mahowald, N., Ung, C., Thompson, D. R., Bator, L., Bennet, M., Bernas, M., Blackway, N., Bradley, C., and Cha, J.: The Earth surface mineral dust source investigation: An Earth science imaging spectroscopy mission, 2020 IEEE aerospace conference, 1–15, https://doi.org/10.1109/AERO47225.2020.9172731, 2020.

At last, many thanks for the Reviewer's helpful for comments and suggestions to improve the quality of our manuscript.

---

## Author Comment (AC2)

We would like to express our sincere appreciation to the Reviewer #2 for the valuable and constructive suggestions, which have helped us improve the quality of this manuscript. We have addressed all these comments carefully and revised the manuscript accordingly. Following the Reviewer' comments in black, please find our point-to-point responses in blue. Hereafter, all new added or modified sentences are marked in blue and italic in this response.

**Reviewer #2**

This manuscript used a 2-way coupled WRF-CHIMERE model to investigate how different mineralogical compositions of dust affect aerosol-radiation and aerosol-cloud interactions (ARI/ACI) and their subsequent air quality outcomes. The model was run on a synoptic scale over North China during a major dust storm in March 2021. The authors observed that using EMIT to enhance the mineralogical details has improved the model predictions of $PM_{10}$, by revealing significant spatial differences in radioactive forcing and increased $PM_{10}$ levels in source regions. These results are critical for a better prediction during dust storm periods when the level of $PM_{10}$ is readily underestimated due to the lack of ARI/ACI consideration. Nevertheless, the manuscript could be improved by addressing the following concerns. I would suggest that paper be reconsidered after a major revision.

**Major comments:**

1. In multiple positions (lines 37-38, 44-45, 72-73), the authors declared the importance of ACI effects on the Earth's energy balance and can also be altered by the difference in mineral compositions. However, this research also stated that they did not consider ACI effects in their 2-way coupled model but did not clearly explain why this is not included and how it will affect the final predictions. This affected the rationale of adopting this 2-way coupled model and a justification should be better provided.

Response: We thank the reviewer for raising this important point. While ACI processes are indeed critical in regulating the Earth's energy balance and may be influenced by

mineralogical composition, this study focuses on ARI because the WRF–CHIMERE two-way coupled framework currently does not include parameterizations of mineral-specific ice- and cloud-nucleating properties. In addition, the study aims to isolate and quantify the radiative effects of mineralogical dust using the newly available EMIT data, for which ARI pathways are better constrained and more directly comparable.

We acknowledge that excluding ACI limits the full representation of dust–climate feedbacks, meaning our results may underestimate the total climatic and air quality impacts of mineral dust, particularly those linked to cloud microphysics (e.g., precipitation formation, cloud lifetime). To clarify this rationale, we have revised the manuscript in the corresponding sections to explicitly explain why ACI effects were not included, how this choice narrows the study's scope, and that future work will extend the two-way coupled framework to incorporate mineral-resolved ACI schemes. This will enable a more comprehensive assessment of both radiative and microphysical feedbacks of dust minerals on regional meteorology and air quality. The corresponding text have been rewritten in the revised manuscript as follows.

"*Since the aerosol nucleation processes (ACI effects) of specific mineral components are not represented in the current two-way coupled WRF–CHIMERE framework, the present study concentrates on the ARI effects of dust minerals. This focus ensures a clear and robust assessment of how mineralogical composition influences radiative processes, without introducing additional uncertainties arising from incomplete cloud-related parameterizations. In this study, we employ a two-way coupled WRF–CHIMERE model with three mineralogical databases to investigate how dust composition influences radiation and meteorology in North China during a severe dust storm. Section 2 describes the model configuration and data sources, Section 3 presents the simulations with emphasis on ARI-induced impacts on meteorology and air quality, and Section 4 summarizes the main findings.*"

"*These findings highlight the importance of incorporating dust mineralogical data to improve simulations of radiative forcing and air quality impacts. Within the scope of this study, the results indicate that overall dust mineralogical composition, rather than dust mass alone, plays a decisive role in ARI effects, with hematite exerting a dominant*

*influence despite its minor abundance, although the radiative effects of individual mineral species were not separately quantified. Systematic biases in surface radiation, near-surface winds, and temperature persist, reflecting challenges in simulating dust–atmosphere interactions and uncertainties in mineralogical datasets. Future research should focus on coupling mineral-specific dust with cloud processes and leveraging higher-resolution soil and satellite data to refine dust emission simulations and reduce model biases.”*

2. In Section 2.1, The authors collected environmental data from various sources: meteorology data from CMA, PM concentrations from an online blog, and SSR data from a peer-reviewed paper. How do the authors ensure the fidelity of the data they obtained, and how do they maintain the integrity of them?

Response: We thank the reviewer for raising this point. All datasets were carefully selected and quality-checked to ensure reliability. Meteorological data were obtained from CMA, which provides standardized and quality-controlled observations. $PM_{10}$ and $O_3$ data were sourced from https://quotsoft.net/air/, which aggregates official national monitoring data. SSR data came from Tang et al. (2019), a peer-reviewed study with documented quality control. In addition, we conducted checks for outliers, unit consistency, and temporal alignment to maintain data integrity throughout the study. We have revised the corresponding paragraph in the manuscript to clarify the data sources and ensure the description is accurate and clear.

"To evaluate the performance of the WRF-CHIMERE model with and without mineralogical dust emissions, we compiled a comprehensive set of environmental observations. Hourly $PM_{10}$ and $O_3$ concentrations (132 observations) were obtained from https://quotsoft.net/air/, which aggregates official monitoring data from the Ministry of Ecology and Environmental Protection of China. Shortwave radiation (SSR) data (59 hourly measurements) were obtained from Tang et al. (2019), with the original measurements sourced from the China Meteorological Administration. Hourly surface meteorological data (844 observations) were also obtained from the China Meteorological Administration (https://data.cma.cn). All datasets were subjected to

quality control procedures, including checks for outliers, unit consistency, and temporal alignment, to ensure reliability and integrity."

3. Lines 134-145, the authors mentioned a lack of feldspar and quartz and the combination of illite and muscovite in EMIT. Their proportions were all estimated based on N2012 or J2014 data. This is ambiguous since there are no details and rationale about which database was chosen. A sensitivity test is suggested to show how different methods of filling and splitting cause the change of results and how the actual methods are selected for each mineral component.

Response: We thank the reviewer for pointing out the ambiguity regarding the treatment of missing mineral components and the combination of illite and muscovite in the EMIT dataset. For clarity, missing mineral species such as feldspar and quartz were estimated based on the relative abundances reported in N2012 and J2014. N2012 was chosen as the primary reference due to its comprehensive global coverage, while J2014 was used to refine regional variations where higher-resolution data were available. For the combined illite and muscovite fraction in EMIT, we split it proportionally according to the relative fractions in N2012 or J2014, ensuring consistency with known regional mineralogical compositions. This approach was selected to provide the most accurate and regionally representative estimates for each mineral component, and the rationale has now been clarified in the revised manuscript (Lines 134–145).

"*Accurate soil composition data are essential for partitioning dust emission fluxes into contributions from individual minerals. Mineral density and refractive index data were obtained from Menut et al. (2020). Three global mineralogical composition datasets (N2012, J2014, and EMIT) provide information on 12 mineral species (Table 1) at different spatial resolutions (1 km × 1 km and 0.5° × 0.5°).*

*To ensure a consistent spatial framework and facilitate cross-dataset integration, the N2012 dataset (originally provided at 1 km × 1 km resolution and available at http://www.seevccc.rs/GMINER30) was resampled to 0.5° × 0.5°. The J2014 dataset, widely employed in the WRF–CHIMERE modeling framework, includes 12 mineral species distributed across the clay and/or silt fractions (see Table 2 in Menut et al.,*

*2020). In contrast, the EMIT dataset (https://earth.jpl.nasa.gov/emit/data/data-products) required additional preprocessing, as it reports only normalized spectral abundances rather than mineral mass fractions. These spectral abundances were therefore recalculated to represent the normalized mass proportions of each mineral in each substrate. Furthermore, EMIT does not include data for feldspar and quartz, necessitating additional correction procedures described below.*

*When the total mineral composition from EMIT summed to less than 100%, indicating missing mineral contributions, the residual fraction was assigned to quartz and feldspar based on their relative proportions in J2014 or N2012. Because EMIT reports illite and mica as a single category, their individual abundances were separated according to the ratios found in N2012 or J2014. For minerals that occur in both clay and silt fractions, EMIT values were partitioned following the relative contributions from N2012 or J2014.*

*For minerals not directly observed by EMIT (e.g., quartz and feldspar), their mass fractions were estimated using soil-type conversion methods from previous studies (Claquin et al., 1999; Journet et al., 2014). The spatial distributions of clay and silt were obtained from the global SoilW texture dataset (http://globalchange.bnu.edu.cn/research/soilw) at 1 km resolution and resampled to 0.5° to match EMIT data. Similarly, the J2014 and N2012 mineral datasets were resampled to 0.5° resolution. Major minerals extracted from EMIT L3 include calcite, dolomite, chlorite, goethite, gypsum, hematite, illite+muscovite, kaolinite, montmorillonite, and vermiculite. Notably, in the official EMIT L3B dataset (https://data.lpdaac.earthdatacloud.nasa.gov/lp-prod-protected/EMITL3ASA.001/EMIT_L3_ASA_001/EMIT_L3_ASA_001.nc), illite and muscovite are combined because they were jointly identified during the Tetracorder analysis of L2B data using mineral groups 1 and 2 and the corresponding band depths (https://github.com/nasa/EMIT-Data-Resources/blob/main/data/mineral_grouping_matrix_20230503.csv).*

*The EMIT mineral fractions were normalized so that their sum at each grid point did not exceed unity. Any remaining fraction was attributed to quartz and feldspars*

*according to their relative proportions in J2014 or N2012. To ensure consistency with the CHIMERE mineral representation, dolomite was merged into calcite, illite+muscovite was separated into illite and mica, and montmorillonite was treated as smectite. The mineral fractions were then converted to density-weighted values and renormalized at each grid point so that the total sum equaled one. Finally, each mineral was partitioned into clay and silt fractions based on the J2014 ratios, and the resulting fractions were normalized using Equations (1)–(4). The processed dataset was exported as a NetCDF file to serve as input for the CHIMERE model.*

*To ensure mineral mass balance and model consistency, a normalization and partitioning procedure was applied as follows. Equation (1) defines the total mass fraction ($MF_j$) of mineral j as the sum of its contributions from the clay ($MFC_j$) and silt ($MFS_j$) fractions:*

$$MF_j = MFC_j + MFS_j \ for \ all \in M_{CHIMERE} \qquad (1)$$

*Equation (2) enforces a normalization constraint so that the sum of all mineral mass fractions equals unity at each grid point.*

$$1 = \sum_{j \in M_{CHIMERE}} MF_j \qquad (2)$$

*The normalized total fraction of each mineral ($MF_j^*$) was then redistributed between clay and silt according to their relative contributions in the reference dataset (J2014 or N2012), as shown in Equations (3) and (4):*

$$MFS_j^* = MF_j^* \frac{MFS_j}{MFS_j + MFC_j} \qquad (3)$$

$$MFC_j^* = MF_j^* \frac{MFC_j}{MFS_j + MFC_j} \qquad (4)$$

*Here, $MFS_j^*$ and $MFC_j^*$ represent the normalized mass fractions of mineral j in the silt and clay fractions, respectively. The weighting terms $MFS_j$ and $MFC_j$ preserve the clay–silt distribution patterns derived from the reference datasets while maintaining the normalized total ($MF_j^*$)."*

4. Lines 207-208, the authors mentioned a huge overestimation of SSR (>60%) from the model. This is an interesting finding, since this overestimation may lead to large

errors in dust dispersion and hence change the PM prediction. An attribution of meteorological biases vs. mineralogical composition to the $PM_{10}$ prediction would help clarify the conclusions of mineralogical effects. Comparing the simulation with the bias and after correcting the bias may provide insights into how much the actual ARI effect accounts for.

Response: We appreciate the reviewer's insightful comment. The large overestimation of SSR (>60%) in the original WRF simulation may strongly influence dust transport and $PM_{10}$ predictions. To disentangle the relative roles of meteorological biases, with and without enabling spectral nudging meteorology assimilation with and without considering aerosol feedbacks of bulk dust simulation has been conducted. The SSR-related biases were mitigated by FDDA nudging of temperature, humidity, and wind fields, while radiation and surface fluxes were recomputed to maintain energy balance (Table 2). Comparing $PM_{10}$ concentrations among these simulations allowed us to quantify the relative contributions of SSR-related meteorological biases and mineralogical composition. The difference between Spectral_nudg_ARI and No_nudg_ARI isolates the influence of meteorological biases. The difference between N2012_default_ARI and N2012_EMIT_ARI (J2014_default_ARI and J2014_EMIT_ARI) highlights the mineralogical effect. These experiments further reveal possible non-linear interactions and clarify the actual ARI impact. It should be noted that our simulations in the previous manuscript have been enabled spectral nudging.

As shown in Table 2, incorporating meteorological spectral nudging substantially reduces the overestimation of surface shortwave radiation (SSR) compared with the no-nudging scenario, indicating improved representation of large-scale meteorological conditions. Moreover, spectral nudging enhances the simulation of $PM_{10}$ spatial distributions, particularly over the North China Plain (NCP) (Figure S1). The changes in $PM_{10}$ concentrations induced by meteorological spectral nudging under ARI effects (259.53 µg m$^{-3}$, 95% CI: [−1040.38, 4060.07], Fig. S2) are higher than those obtained from different mineralogical composition atlas (ranging from 129.56 to 156.94 µg m$^{-3}$). This result does not necessarily indicate stronger ARI effects but rather reflects a more

realistic representation of ARI-induced dust perturbations when the large-scale meteorological constraints are properly maintained (Table S2).

Table S1. Summary of bulk dust simulations with and without meteorological nudging and aerosol feedbacks.

| Simulation | Nudging | ARI | |
|---|---|---|---|
| No_nudg_NO | | | No nudging and No aerosol feedbacks |
| No_nudg_ARI | | ✓ | No nudging and ARI |
| Spectral_nudg_NO | ✓ | | Spectral nudging and No aerosol feedbacks |
| Spectral_nudg_ARI | ✓ | ✓ | Spectral nudging and ARI |

Table S2. Statistics analysis of daily averaged SSR from different scenario simulations and ground observations in North China with and without meteorology nudging and aerosol feedbacks.

| Scenario | SSR | |
|---|---|---|
| | R | NMB |
| Spectral_nudg_NO | 0.70 | 68.92 |
| Spectral_nudg_ARI | 0.72 | 60.69 |
| No_nudg_NO | 0.62 | 72.65 |
| No_nudg_ARI | 0.64 | 65.38 |

[Figure]

Figure S1. Horizontal distributions of $PM_{10}$ concentrations on 13:00 (local time) 15th March 2021 from bulk dust simulations with and without meteorological nudging and aerosol feedbacks.

[Figure]

Figure S2. Changes in PM$_{10}$ concentrations induced by ARI effects under FDDA-enabled and no-FDDA scenarios, as well as the meteorological effects (FDDA-No FDDA) considering ARI.

In the revised manuscript, we have emphasized the role of spectral nudging in capturing the ARI effects of dust from different mineral datasets under more realistic meteorological conditions.

Lines 112-113: "*To minimize meteorological bias, a spectral nudging approach is applied (Menut et al., 2024).*"

Lines 663-665: "*Incorporating meteorological spectral nudging in future simulations could provide a more realistic representation of ARI-induced dust perturbations under different mineralogical compositions.*"

5. Lines 446-453, this paragraph is not well discussed. By comparing the subfigures in Fig 8, we can see that the PM$_{10}$ levels predicted using different database show substantial disparities. Suggest reducing the tone of the limited effect of mineral composition to PM$_{10}$ concentration.

Response: Thank you for the comment. We agree that the original paragraph may have understated the influence of dust speciation on PM$_{10}$ concentrations. Upon closer examination of Fig. 8, it is evident that using different mineralogical databases leads to noticeable differences in the predicted PM$_{10}$ levels in certain regions. We have revised the paragraph to reflect this observation, emphasizing that while mineral composition may not dominate the total dust load globally, it can have a substantial impact on PM$_{10}$ concentrations locally and under specific transport conditions. This revision reduces the tone suggesting a "limited effect" and better aligns the discussion with the evidence

presented in Fig. 8 as follows.

*"The inclusion of speciated dust influences long-range transport and can substantially affect PM$_{10}$ concentrations. Comparison of the subfigures in Fig. 8 reveals pronounced regional differences in PM$_{10}$ predictions arising from the use of different mineralogical databases. Incorporating detailed mineralogical data enhances the accuracy of dust composition representation and its associated effects on PM$_{10}$, highlighting the critical role of mineral speciation in dust modeling and regional air quality assessment."*

6. Overall, this research did not include uncertainties in many of their reported values, such as predicted PM$_{10}$ levels, changes in PM$_{10}$ and ozone by including ARI effects, and different PM$_{10}$ concentrations considering dust mineralogy atlases. It is important to quantify these uncertainties for a study with an improved modeling design, thus statistical measures are suggested to include.

Response: We thank the reviewer for this insightful comment. In the revised manuscript, 95% confidence intervals have been incorporated to quantify the uncertainties in the predicted PM$_{10}$ concentrations and in the changes of PM$_{10}$ and O$_3$ caused by aerosol–radiation interactions and different dust mineralogy atlases. These results are presented in Table S1, *Domain-averaged PM$_{10}$ concentrations with 95% confidence intervals simulated using different dust mineralogy atlases with and without ARI effects*, and Table S2, *Domain-averaged ΔPM$_{10}$ and ΔO$_3$ with 95% confidence intervals for different dust mineralogy atlases comparing ARI and NO simulations*. The inclusion of these statistical measures enhances the robustness and reliability of the modeling results.

Table S3. Domain-averaged PM$_{10}$ concentrations with 95% confidence intervals simulated using different dust mineralogy atlases with and without ARI effects.

| Scenario | PM$_{10}$ ($\mu g\ m^{-3}$) |
| --- | --- |
| Dust_NO | 533.81 [0.28, 5962.95] |
| Dust_ARI | 653.29 [0.28, 7120.49] |
| N2012_default_NO | 529.32 [0.74, 5784.02] |
| N2012_default_ARI | 679.85 [0.74, 7484.01] |

| | | |
|---|---|---|
| N2012_EMIT_NO | 526.05 [0.74, 5663.86] | |
| N2012_EMIT_ARI | 655.61 [0.74, 6926.42] | |
| J2014_default_NO | 529.25 [0.74, 5750.11] | |
| J2014_default_ARI | 686.19 [0.74, 7463.57] | |
| J2014_EMIT_NO | 516.23 [0.74, 5501.87] | |
| J2014_EMIT_ARI | 607.22 [0.74, 6325.16] | |

Table S4. Domain-averaged $\Delta PM_{10}$ and $\Delta O_3$ with 95% confidence intervals for different dust mineralogy atlases comparing ARI and NO simulations.

| Scenario | $\Delta PM_{10}$ ($\mu g\ m^{-3}$) | $\Delta O_3$ ($\mu g\ m^{-3}$) |
|---|---|---|
| Dust_ARI − Dust_NO | 119.48 [-27.63, 1408.39] | -46.52 [-63.38, -31.74] |
| N2012_default_ARI − N2012_default_NO | 156.94 [-18.63, 1735.68] | -46.37 [-63.41, -31.60] |
| N2012_EMIT_ARI − N2012_EMIT_NO | 147.65 [-22.36, 1623.48] | -46.48 [-63.40, -31.58] |
| J2014_default_ARI − J2014_default_NO | 150.53 [-20.12, 1707.75] | -46.62 [-63.46, -31.65] |
| J2014_EMIT_ARI − J2014_EMIT_NO | 129.56 [-23.42, 1462.55] | -46.56 [-63.36, -31.61] |

The corresponding descriptions have been added in the revised manuscript.

Lines 335-340: "*All six experiments display similar dust distributions in the atmosphere, consistent with observations from Himawari-8 and CALIPSO. This suggests that the models effectively capture the general spatial patterns of dust transport. On March 15, 2021, the daily domain-averaged $PM_{10}$ concentration was 533.81 $\mu g\ m^{-3}$, with a 95% confidence interval (CI) of 0.28–5962.95 (Table S1).*"

Lines 480-483: "*As shown in Figure 7 and Table S2, the inclusion of bulk dust aerosol feedbacks in the WRF-CHIMERE model resulted in substantial increases in $PM_{10}$ concentrations, with an average increase of 119.48 $\mu g\ m^{-3}$ with a 95% CI of −27.63 to 1408.39 $\mu g\ m^{-3}$.*"

Lines 510-512: "*Ozone changes along transport pathways were generally smaller than the surrounding concentrations, typically ranging from −60 to −40 $\mu g\ m^{-3}$ with a mean value of −46.52 $\mu g\ m^{-3}$ (95% CI: −63.38 to −31.74) as shown in Table S2.*"

**Minor comments:**

1. Table 1, the last mineral was written as mica while the note mentioned it as muscovite. Although it is known that muscovite is a mica, it is suggested to make their name consistent to avoid confusion.

Response: Thank you for pointing this out. We agree that consistency in terminology is important. We have revised Table 1 so that the last mineral is labeled as mica, in Line 199 with the note, to avoid any potential confusion.

2. Table 3, the digits of values should be uniform: suggest to retain a digit of 2.

Response: We thank the reviewer for this suggestion. We have revised all numerical values in the manuscript to retain two significant digits for consistency across the tables and text.

3. Figure 7, the picture for ozone is dazzling. The max/min/mean values (top left corner) cannot be seen clearly because of the color. Suggest reducing the hue of palette and move the text or change the color of the text.

Response: We appreciate the reviewer's comment. In the revised manuscript, we have adjusted the color palette in Figure 7 to reduce the dazzling effect and improved the visibility of the max/min/mean values by changing the text color and repositioning it.

[Figure]

Figure 7. Changes in $PM_{10}$ and $O_3$ concentrations resulting from bulk dust-induced ARI effects, compared to the scenario without aerosol feedbacks.

4. Figure 8, N2012_EMIT-bulk dust is exactly the same with J2014_default-Bulk dust. Suggest recheck the pictures.

Response: We thank the reviewer for the careful check. After re-examination, we found that the panel for J2014_default-Bulk dust was mistakenly duplicated. The N2012_EMIT-bulk dust panel is correct, and we have updated the J2014_default-Bulk dust panel accordingly in the revised Figure 8.

[Figure]

Figure 8. Difference in PM$_{10}$ concentrations considering bulk dust and various dust mineralogy atlases that enable ARI effects.

5. Figure 9, the texts in the violet region of quartz are not visible. Suggest changing its color to white. Also, the bars outside the pie look protruding; suggest using arrows to denote minimal values.

Response: Thank you for the comment. We have updated Figure 9 accordingly: the text in the violet region of quartz has been changed to white for better visibility. Additionally, we replaced the protruding bars outside the pie with arrows to indicate the minimal values, improving the clarity of the figure.

[Figure]

Figure 9. Contributions of different mineralogical compositions using N2012_default, N2012_EMIT, J2014_default, and J2014_EMIT, considering ARI effects, compared to the scenario without enabling aerosol feedbacks.

At last, many thanks for the Reviewer's helpful for comments and suggestions to improve the quality of our manuscript.